# A POINCARÉ INEQUALITY AND CONSISTENCY RESULTS FOR SIGNAL SAMPLING ON LARGE GRAPHS

**Thien Le**
MIT
thienle@mit.edu

**Luana Ruiz**
Johns Hopkins University
lrubini1@jh.edu

**Stefanie Jegelka**
TU Munich, MIT
stefje@mit.edu

## ABSTRACT

Large-scale graph machine learning is challenging as the complexity of learning models scales with the graph size. Subsampling the graph is a viable alternative, but sampling on graphs is nontrivial as graphs are non-Euclidean. Existing graph sampling techniques require not only computing the spectra of large matrices but also repeating these computations when the graph changes, e.g., grows. In this paper, we introduce a signal sampling theory for a type of graph limit—the graphon. We prove a Poincaré inequality for graphon signals and show that complements of node subsets satisfying this inequality are unique sampling sets for Paley-Wiener spaces of graphon signals. Exploiting connections with spectral clustering and Gaussian elimination, we prove that such sampling sets are consistent in the sense that unique sampling sets on a convergent graph sequence converge to unique sampling sets on the graphon. We then propose a related graphon signal sampling algorithm for large graphs, and demonstrate its good empirical performance on graph machine learning tasks.

## 1    INTRODUCTION

Graphs are ubiquitous data structures in modern data science and machine learning. Examples range from social networks (Kempe et al., 2003; Barabási et al., 2000) and recommender systems (Ying et al., 2018) to drug interactions (Zitnik et al., 2018) and protein folding (Jumper et al., 2021), in which the graph can have tens of thousands to millions of nodes and edges (Takac & Zábovský, 2012). The ability to sense systems at this scale presents unprecedented opportunities for scientific and technological advancement. However, it also poses challenges, as traditional algorithms and models may need to scale more efficiently to large graphs, including neural graph learning methods.

However, the large size of modern graphs does not necessarily indicate the degree of complexity of the problem. In fact, many graph-based problems have low intrinsic dimensions. For instance, the 'small-world phenomenon' (Kleinberg, 2000) observes that any two entities in a network are likely connected by a short sequence of intermediate nodes. Another example are power-law graphs, where there are few highly connected influencers and many scattered nodes (Barabási et al., 2000).

At a high level, this paper studies how to exploit these simplicities in large graphs to design scalable algorithms with theoretical guarantees. In particular, we combine two ideas: graph limits, which are used to approximate large, random graphs; and sampling theory, which studies the problem of representing (graph) signals using the smallest possible subset of data points (nodes), with the least possible loss of information. We then illustrate how to use the resulting sampling techniques to compress graphs for GNN training and to compute faster, subsampled positional encodings.

**Graphons and graph limits.** Leveraging continuous limits to analyze large discrete data is helpful because limits often reveal the intrinsic dimension of the data. E.g., in Euclidean domain, the Fourier transform (FT) of a continuous signal is easier to analyze than the FT of its discrete counterpart, which is periodic and may exhibit aliasing. We propose to study the graph signal sampling problem on a graph limit called graphon. Graphons can be thought of as undirected graphs with an uncountable number of nodes, and are both random graph models and limits of large dense graphs (Borgs et al., 2008; Lovász, 2012).

**(Graph) signal sampling.** Sampling theory is a long-standing line of work with deep roots in signal processing. Traditionally, sampling seeks to answer the fundamental question: if one can only observe discrete samples of an analog (continuous) signal, under what conditions can the analog signal be perfectly reconstructed? On a graph on $n$ nodes, signals are vectors $\mathbf{x} \in \mathbb{R}^n$ that map each node to some value. The graph signal sampling problem is then defined as follows.

**Problem 1.** *For some signal space $\mathcal{X}$ of interest, find subsets $S$ of nodes such that if $\mathbf{x}, \mathbf{x}' \in \mathcal{X}$ and $x_i = x'_i$ for all $i \in S$ then $x_j = x'_j$ for all other nodes. Thus, such a set can uniquely represent any signals in $\mathcal{X}$ and is called a uniqueness set for $\mathcal{X}$.*

Problem 1 was first studied by Pesenson (2008), who introduced Paley-Wiener (PW) spaces for graph signals, defined graph uniqueness sets, and derived a Poincaré inequality for discrete graph signals that allows recovering such uniqueness sets. These definitions are reviewed in Section 2. Graph signal sampling theory subsequently found applications in the field of graph signal processing (GSP) (Shuman et al., 2013; Ortega et al., 2018), with Chen et al. (2015) describing how sampling sets can be obtained via column-wise Gaussian elimination of the eigenvector matrix.

**Current limitations.** Though widely used, Chen et al. (2015)'s approach requires expensive spectral computations. Several methods, briefly discussed in Section 1.1, have been proposed to circumvent these computations; however, these approaches still present stringent tradeoffs between complexity and quality of approximation on very large graphs. Perhaps more limiting, the discrete sampling sets yielded by these methods are no longer applicable if the graph changes, as often happens in large real-world network problems, e.g., an influx of new users in a social network.

**Contributions.** To address the abovementioned issues, we propose sampling uniqueness sets on the limit graphon. By solving a single sampling problem at the graph limit (graphon), we obtain a uniqueness set that generalizes to any large finite graphs in a sequence converging to the limit graphon. We provide both theoretical guarantees and experiments to verify this generalization in downstream graph-based tasks. In summary, our contributions are:

1. Motivated by Pesenson (2008), we formulate signal sampling over a graphon and study traditional sampling theory notions such as Paley-Wiener spaces and uniqueness sets[1] in a Euclidean setting of $L^2([0, 1])$ while still incorporating graph structure into the sampling procedure.

2. We prove a Poincaré inequality for graphons and relate bandlimitedness in graphon signal space to optimal sampling sets[1]. This generalizes previous results on finite graphs and rigorously answers a reconstruction question. Unlike other results for graphon signal processing in the literature, we do not require any continuity or smoothness assumption on the graphon.

3. We uncover a connection between graphon sampling and kernel spectral clustering and design a Gaussian-elimination-based algorithm to sample from the graphon uniqueness set with provable consistency, using an argument from (Schiebinger et al., 2015).

4. We empirically evaluate our sampling method on two tasks: (1) transferability: training a GNN on subsampled graphs and testing on the full graph; (2) accelerating the computation of positional encodings for GNNs by restricting them to a sampled subset of nodes.

## 1.1 RELATED WORK

**Graphons in machine learning.** In machine learning, graphons have been used for network model estimation (Borgs et al., 2015), hierarchical clustering (Eldridge et al., 2016) and to study the theoretical properties of graph neural networks (GNNs) on large graphs. Specifically, Ruiz et al. (2020b) have shown that graph convolutions converge to graphon convolutions, further proving a non-asymptotic result that implies that GNNs are transferable across graphon-sampled graphs (Ruiz et al., 2020a). Similar studies have been done using graphops (Le & Jegelka, 2023), which are very general graph limits that range from graphons to very sparse graphs. Graphons have also been used to show convergence of GNN training on increasing graph sequences (Cervino et al., 2023), to prove PAC-Bayes bounds for GNN learning (Maskey et al., 2022), and to study the learning dynamics of wide large-graph NNs (Krishnagopal & Ruiz, 2023).

---

[1]Similar sampling theory was also concurrently developed by Parada-Mayorga & Ribeiro (2024).

**Graph signal sampling.** Graph signal sampling has been studied at length in GSP. Chen et al. (2015) describe how sampling sets can be obtained via column-wise Gaussian elimination of the eigenvector matrix and derive conditions for perfect reconstruction. Noting that this approach requires expensive spectral computations, several methods were proposed to avoid them. E.g., Anis et al. (2016) calculate eigenvalue and eigenvector approximations using power iteration; Marques et al. (2015) compute $n$ signal aggregations at a single node $i$ to construct an $n$-dimensional local signal from which $K$ elements are sampled; and Chamon & Ribeiro (2017) do greedy sampling and provide near optimal guarantees when the interpolation error is approximately supermodular.

**Connections with other sampling techniques.** The sampling algorithm we propose is based on a greedy iterative procedure that attempts to find the signal with the lowest total variation on the complement of the current sampling set $S$, and adds the node corresponding to the largest component in this signal to $S$. This heuristic is derived by trying to maximize the largest eigenvalue of the normalized Laplacian restricted to $S$ (see (Anis et al., 2016, Section IV.C) for a detailed discussion). Thus, our algorithm has close connections with E-optimal design, which minimizes the largest eigenvalue of the pseudo-inverse of the sampled matrix (Pukelsheim, 2006), and with dual volume sampling (Avron & Boutsidis, 2013; Li et al., 2017), which provides approximation guarantees for E-optimal sampling. This type of objective also appears in effective resistance/leverage scores sampling (Ma et al., 2014; Rudi et al., 2018), which is used for graph sparsification (Spielman & Srivastava, 2008).

Recent work by Parada-Mayorga & Ribeiro (2024), concurrent with ours, also generalized PW spaces and uniqueness sets to graphons. Similarly, they proved a Poincaré inequality and proposed a sampling algorithm for graphon signals. Their main results quantitatively compare the Poincaré constant across different graphon-signal spaces, implying convergence of this constant for a convergent graph sequence, unlike our work, which analyzes consistency of sampling via spectral clustering.

## 2 Preliminaries

### 2.1 Graph Signal Processing

**Setup.** We consider graphs $\mathbf{G} = (\mathcal{V}, \mathcal{E})$ with $n$ nodes and edges $\mathcal{E} \subseteq \mathcal{V} \times \mathcal{V}$. We write a graph's adjacency matrix as $\mathbf{A} \in \mathbb{R}^{n \times n}$; its degree matrix as $\mathbf{D} = \mathrm{diag}(\mathbf{A1})$; and its Laplacian matrix as $\mathbf{D} - \mathbf{A}$. We also consider the normalized adjacency and Laplacian matrices $\bar{\mathbf{A}} = (\mathbf{D}^\dagger)^{1/2} \mathbf{A} (\mathbf{D}^\dagger)^{1/2}$ and $\bar{\mathbf{L}} = \mathbf{I} - \bar{\mathbf{A}}$ (where $\cdot^\dagger$ is the pseudoinverse), with eigendecomposition $\bar{\mathbf{L}} = \mathbf{V} \mathbf{\Lambda} \mathbf{V}^T$ and eigenvalues $\lambda_1 \leq \ldots \leq \lambda_n$. We further consider node signals $\mathbf{x} \in \mathbb{R}^n$, which assign data value $x_i$ to node $i$; e.g., in a social network, $x_i$ may represent the political affiliation of person $i$.

**Total variation and graph frequencies.** The total variation of a graph signal is defined as $\mathrm{TV}(\mathbf{x}) = \mathbf{x}^T \bar{\mathbf{L}} \mathbf{x}$ (Anis et al., 2016; Sandryhaila & Moura, 2014). This allows interpreting the eigenvalues $\lambda_i$ as the graph's essential frequencies, with oscillation modes given by the eigenvectors $\mathbf{v}_i = [\mathbf{V}]_{:i}$.

**Graph FT and Paley-Wiener spaces.** We may analyze signals on the graph frequency domain via the graph Fourier transform (GFT). The GFT $\hat{\mathbf{x}}$ of $\mathbf{x}$ is its projection onto the Laplacian eigenbasis $\hat{\mathbf{x}} = \mathbf{V}^T \mathbf{x}$ (Sandryhaila & Moura, 2014). The GFT further allows defining bandlimited graph signals, or, more formally, Paley-Wiener (PW) spaces. On $\mathbf{G}$, the PW space with cutoff frequency $\lambda$ is defined as $PW_\lambda(\mathbf{G}) = \{\mathbf{x} \text{ s.t. } [\hat{\mathbf{x}}]_i = 0 \text{ for all } \lambda_i > \lambda\}$ (Anis et al., 2016; Pesenson, 2008).

**Uniqueness sets.** When $\mathcal{X}$ is a PW space $PW_\lambda(\mathbf{G})$ with $\lambda \leq \lambda_K$ for some $K < n$, there exists a subset of at most $K$ nodes that perfectly determine any signal in $\mathcal{X}$ called uniqueness set. The following theorem from (Anis et al., 2016) gives conditions under which a proposed subset $\mathcal{S}$ is a uniqueness set for $PW_\lambda(\mathbf{G})$.

**Theorem 1** (Uniqueness sets for $PW_\lambda(\mathbf{G})$). *Let $\mathcal{S} \subseteq \mathcal{V}$. Let $\mathbf{V}_K \in \mathbb{R}^{n \times K}$ denote the first $K$ columns of the eigenvector matrix $\mathbf{V}$ and $\mathbf{\Psi}_\mathcal{S} \in \mathbb{R}^{K \times K}$ be the submatrix of $\mathbf{V}$ with rows indexed by $\mathcal{S}$. If $\mathrm{rank}\mathbf{\Psi}_\mathcal{S} = K$, then $\mathcal{S}$ is a uniqueness set for $PW_\lambda(\mathbf{G})$ for all $\lambda \leq \lambda_K(\mathbf{G})$. If $\lambda_K \leq \lambda < \lambda_{K+1}$ then $\mathrm{rank}\mathbf{\Psi}_\mathcal{S} = K$ is also necessary.*

In addition to providing a sufficient condition to verify if a set is a uniqueness set for some PW space, this theorem suggests a two-step strategy for obtaining such sets: first compute $\mathbf{V}_K$, and then design a sampling method that outputs $\mathcal{S}$ such that $\mathrm{rank}\mathbf{\Psi}_\mathcal{S} = K$. However, these sampling strategies,

e.g., the one suggested by Thm. 1, can be limiting on large graphs as they require computing the eigendecomposition of a large matrix.

## 2.2 GRAPHON SIGNAL PROCESSING

**Graphons and graphon signals.** A underline{graphon} is a symmetric, bounded, measurable function $\mathbf{W} : \Omega \times \Omega \to [0, 1]$, where $\Omega$ is a general measurable space (Borgs & Chayes, 2017). We assume that there exists an invertible map $\beta : \Omega \to [0, 1]$ and w.l.o.g.,we can also write $\mathbf{W} : [0, 1]^2 \to [0, 1]$. Graphons are only defined up to a bijective measure-preserving map, similar to how finite graphs are defined up to node permutations. Graphons are limits of graph sequences $\{\mathbf{G}_n\}$ in the so-called homomorphism density sense (Borgs et al., 2008), and can also be seen as random graph models where nodes $u_i, u_j$ are sampled from $\Omega$ and edges $(u_i, u_j) \sim \text{Bernoulli}(\mathbf{W}(u_i, u_j))$. Graphons can also be motivated via infinite exchangeable graphs (Hoover, 1979; Aldous, 1981).

underline{Graphon signals} are functions $X : [0, 1] \to \mathbb{R}$. They represent data on the "nodes" of a graphon, i.e., $X(u)$ is the value of the signal at node $u \in [0, 1]$ (Ruiz et al., 2021). Since two graphons that differ on a set of Lebesgue measure 0 are identified, so are graphon signals. We restrict attention to finite-energy signals $X \in L^2([0, 1])$.

**Graphon Laplacian and FT.** Given a graphon $\mathbf{W}$, its underline{degree function} is $\mathbf{d}(v) = \int_0^1 \mathbf{W}(u, v)\mathrm{d}u$. Define the normalized graphon $\bar{\mathbf{W}}(u, v) = \mathbf{W}(u, v)/\sqrt{\mathbf{d}(u)\mathbf{d}(v)}$ if $\mathbf{d}(u), \mathbf{d}(v) \neq 0$ and 0 otherwise. Given a graphon signal $X$, we define the underline{normalized graphon Laplacian}:

$$\bar{\mathcal{L}}X = X - \int_0^1 \bar{\mathbf{W}}(u, \cdot)X(u)\mathrm{d}u. \tag{1}$$

The underline{spectrum} of $\bar{\mathcal{L}}$ consists of at most countably many nonnegative eigenvalues with finite multiplicity in $[0, 2]$. Its essential spectrum consists of at most one point $\{1\}$, and this is also the only possible accumulation point. We enumerate the eigenvalues as $0 \leq \lambda_1 \leq \lambda_2 \leq \ldots \leq 2$. The corresponding set of eigenfunctions $\{\varphi_i\}_{i \in \mathbb{Z}\setminus\{0\}}$ forms an orthonormal basis of $L^2([0, 1])$; see App. B.

We define the graphon Fourier transform (WFT) of signal $X$ as the projection

$$\hat{X}(\lambda_i) = \int_0^1 X(u)\varphi_i(u)\mathrm{d}u \tag{2}$$

for all $i$. Note that this is different from the WFT defined in (Ruiz et al., 2020b), which corresponds to projections onto the eigenbasis of a different but related linear operator.

## 3 SAMPLING THEORY FOR GRAPHONS

We generalize the graph sampling problem studied in Pesenson (2008) to a graphon sampling problem. The sampling procedure returns a (Lebesgue) measurable subset $U \subseteq [0, 1]$. Intuitively, we would like to choose a set $U$ such that sampling from $U$ gives us the most information about the whole signal over $[0, 1]$. These are called uniqueness sets. Similar to finite graphs, when the graphon signals have limited bandwidth, there exist nontrivial (other than $U = [0, 1]$) uniqueness sets. Finding these sets is the main focus of the sampling theory for graphons that we develop here.

For an arbitrary bandwidth cutoff $\lambda > 0$, we use the normalized graphon Laplacian (1) with eigenvalues $0 \leq \lambda_1 \leq \lambda_2 \leq \ldots \leq \lambda_{-2} \leq \lambda_{-1} \leq 2$. First, we define the Paley-Wiener space:

**Definition 1** (Graphon signal $PW_\lambda(\mathbf{W})$ space). *The Paley-Wiener space associated with $\lambda \in [0, 1]$ and graphon $\mathbf{W}$, denoted $PW_\lambda(\mathbf{W})$, is the space of graphon signals $X : [0, 1] \to \mathbb{R}$ such that $\hat{X}(\lambda_i) = 0$ for all $\lambda_i > \lambda$, where $\hat{X}$ is the projection operator defined in Eq. (2).*

The definition of $PW_\lambda(\mathbf{W})$ depends on the underlying limit graphon through the projection operator (2), in particular the positions of its Laplacian eigenvalues. When $\lambda \geq \lambda_{-1}$, $PW_\lambda$ is all of $L^2([0, 1])$ as the definition above is vacuously satisfied. Decreasing $\lambda$ induces some constraints on what functions are allowed in $PW_\lambda$. At $\lambda = 0$, $PW_0 = \{0\}$ contains only the trivial function.

For any signal space $\mathcal{H} \subseteq L^2([0, 1])$, we further define graphon uniqueness sets:

**Definition 2** (Graphon uniqueness set). *A measurable $U \subseteq [0, 1]$ is a uniqueness set for the signal space $\mathcal{H} \subseteq L^2([0, 1])$ if, for any $X, Y \in \mathcal{H}$, $\int_U |X(u) - Y(u)|^2 du = 0$ implies $\|X - Y\|_{L^2([0,1])}^2 = 0$.*

Since $U = [0, 1]$ is a trivial uniqueness set for any $\mathcal{H} \subseteq L^2([0, 1])$, we are mainly interested in the interplay between the bandwidth cutoff $\lambda$ in $PW_\lambda(\mathbf{W})$, and its corresponding non-trivial uniqueness sets. More precisely, we study the question:

**Problem 2.** *Assume that a graphon signal comes from $PW_\lambda(\mathbf{W})$ for some $\lambda$ and $\mathbf{W}$. Is there an algorithm that outputs a uniqueness set $U(\lambda, \mathbf{W})$?*

We answer this question in the positive and provide two approaches. First, by generalizing results by Pesenson (2008) for finite graphs, we give a graphon Poincaré inequality (Thm. 2) for nontrivial measurable subsets of $[0, 1]$. Then, in Thm. 3, we show that if a set $S$ satisfies the Poincaré inequality with constant $\Lambda > 0$ then the complement $U = [0, 1]\backslash S$ is a uniqueness set for $PW_{1/\Lambda}(\mathbf{W})$ (Thm. 3). Thus, we can find uniqueness set $U$ by first finding an $S$ that satisfies the Poincaré inequality with constant $1/\lambda$.

The second approach is more direct: the analogous question for finite graphs admits a straightforward answer using Gaussian elimination (see the discussion underneath Thm. 1). However, in the limit of infinitely many nodes, it does not make sense to perform Gaussian elimination as is. Instead, we form a sequence of graphs $\{\mathbf{G}_n\}$ that converges to the prescribed graphon $\mathbf{W}$. We then prove, using techniques from (Schiebinger et al., 2015), that performing Gaussian elimination with proper pivoting for $\mathbf{G}_n$ recovers sets that converge to a uniqueness set for $PW_\lambda(\mathbf{W})$ (Prop. 5). Finally, we implement and analyze this approach empirically in Section 6.

## 4 MAIN RESULTS

### 4.1 POINCARÉ INEQUALITY AND BANDWIDTH OF UNIQUENESS SET

We start with the first approach to Problem 2, proving a Poincaré inequality for subsets $S \subset [0, 1]$ and showing that this Poincaré inequality implies uniqueness of $[0, 1]\backslash S$ at some bandwidth.

First, we need some definitions. These definitions generalize Pesenson (2008)'s observation that for finite graphs, any strict subset $T$ of the vertex set satisfies a Poincaré inequality with constant determined by spectral properties of another graph $\Gamma(T)$. Intuitively, $\Gamma(T)$ is designed to capture the non-Euclidean geometry induced by nodes in $T$ and their neighbors. We now want to construct an analogous $\Gamma(S)$ in the graphon case. Fix an arbitrary graphon $\mathbf{W}$ and measurable subset $S \subset [0, 1]$. Define the neighborhood $\mathcal{N}(S)$ of $S$ as the measurable set $\mathcal{N}(S) := \left\{v \in [0, 1]\backslash S : \int_S \mathbf{W}(u, v)\mathrm{d}u > 0\right\}$.

To define $\Gamma(S)$, make a copy of $S$ by letting $S'$ be a set disjoint from $[0, 1]$ such that there is a measure-preserving bijection $\theta : S' \to S$. Let $\tilde{S} := S \cup \mathcal{N}(S)$ and $\tilde{S}' := S' \cup \mathcal{N}(S)$. Observe that one can extend $\theta : \tilde{S}' \to \tilde{S}$ by mapping elements of $\mathcal{N}(S)$ to itself. We will define a graphon on the extended domain $D = \tilde{S} \cup S'$:

$$\Gamma(S) : D^2 \to [0, 1] : (u, v) \mapsto \begin{cases} \mathbf{W}(u, v) & \text{if } u \in \tilde{S} \text{ and } v \in \tilde{S} \\ \mathbf{W}(\theta(u), \theta(v)) & \text{if } u \in \tilde{S}' \text{ and } v \in \tilde{S}' \\ 0 & \text{otherwise.} \end{cases} \tag{3}$$

Spectral properties of $\Gamma(S)$ determine the constant in our Poincaré inequality: a class of important results in functional analysis that control the action of the functional (normalized Laplacian) by the (non-Euclidean) geometry of the underlying space (here, a graph).

**Theorem 2** (Graphon Poincaré inequality). *Let $S \subsetneq [0, 1]$ such that $\mathcal{N}(S)$ has positive Lebesgue measure. Denote by $\lambda_1$ the smallest nonzero eigenvalue of the scaled normalized Laplacian operator applied to $\Gamma(S)$. Then for every $X \in L^2([0, 1])$ supported only on $S$, $\|X\|_{L^2} \leq \frac{1}{\lambda_1}\|\overline{\mathcal{L}}X\|_{L^2}$.*

The proof of this theorem is in App. C and generalizes that in (Pesenson, 2008). Next, we prove that if we can find a set $S$ that satisfies a Poincaré inequality with constant $\Lambda$, then its complement is a uniqueness set for any $PW_\lambda(\mathbf{W})$ with $\lambda < 1/\Lambda$.

**Theorem 3.** *Let $S$ be a proper subset of $[0, 1]$ satisfying the Poincaré inequality*

$$\|X\|_{L^2} \leq \Lambda\|\overline{\mathcal{L}}X\|_{L^2} \tag{4}$$

*for all $X \in L^2([0, 1])$ supported only on $S$. Then, $U = [0, 1]\backslash S$ is a uniqueness set for any $PW_\lambda(\mathbf{W})$ with $\lambda < 1/\Lambda$.*

The proof of this result is in App. C, providing an answer to Problem 2: given a bandwidth limit $\lambda$, one can find a uniqueness set $U$ by searching through measurable sets $S$ and computing the smallest nonzero eigenvalue $\lambda_1$ of $\Gamma(S)$. If $\lambda < \lambda_1$ then $U = [0,1] \backslash S$ is a uniqueness set. This approach is inefficient as we may need to check every $S$. Next, we investigate a more efficient approach.

## 4.2 Gaussian elimination and convergence of uniqueness sets

Our second approach to Problem 2 relies on approximating the graphon with a sequence of graphs $\{\mathbf{G}_n\}$ which has the graphon as its limit, and solving Problem 2 in one of these graphs. While attempting to solve the graphon sampling problem on a finite graph may appear tautological, our goal is to exploit the countable (and often finite) rank of the graphon to make the problem tractable.

To establish the connection between the continuous sampling sets in a graphon and its finite rank $K$, we partition the graphon sampling set into $K$ elements and view each element as representing a mixture component or "cluster". This leads to a connection to mixture models and spectral clustering, which we exploit in two ways. First, to quantify the quality of the graphon sampling sets via a "difficulty" function borrowed from (Schiebinger et al., 2015) relating to the separability of the mixture components. Second, similar to consistency of kernelized spectral clustering, to prove that in convergent graph sequences, graph sampling sets converge to graphon sampling sets.

**Graphons are equivalent to mixture models of random graphs.** To make the above connection rigorous, the first step is to show we can view the graphon as a mixture model of random graphs.

**Definition 3** (Mixture model for random graphs). *Let $\Omega \subset \mathbb{R}^d$ be a compact space and $\mathcal{P}(\Omega)$ the space of probability measures on $\Omega$. For some number of components $K$, components $\{\mathbb{P}_i \in \mathcal{P}(\Omega)\}_{i=1}^K$, weights $\{w_i \geq 0\}_{i=1}^K$ that sum to 1, and a bounded, symmetric, measurable kernel $\mathbf{k} : \Omega \times \Omega \to [0,1]$, a mixture model for random graphs $\mathbb{K}(\Omega, \mathbb{P}, \mathbf{k})$ samples nodes from some mixture distribution; then sample edges using $\mathcal{B}$ - the Bernoulli distribution over the kernel $\mathbf{k}$:*

$$\omega_w \sim \mathbb{P} := \sum_{i=1}^K w_i \mathbb{P}_i, \text{ for } 1 \leq w \leq n, \qquad (u,v) \sim \mathcal{B}(\mathbf{k}(\omega_u, \omega_v)), \text{ for } 1 \leq u, v \leq n. \quad (5)$$

Historically, some authors (Borgs & Chayes, 2017) have defined graphons as in Def. 3, where $\mathbb{P}$ is not necessarily a mixture. Under mild conditions on $\mathbb{P}$, we assert that our simpler definition of a graphon is still equivalent to a random graph model. We leave the proof to App. D.

**Proposition 1.** *Assume the CDF of $\mathbb{P}$ is strictly monotone. Then, the mixture model $\mathbb{K}(\Omega, \mathbb{P}, \mathbf{k})$ (Def. 3) is equivalent to the random graph model $\mathbb{W}([0,1], \mathbb{U}, \mathbf{W})$ where $\mathbf{W} : [0,1]^2 \to [0,1]$ is a graphon given by $\mathbf{W} = \mathbf{k} \circ \beta$ and $\beta : [0,1] \to \Omega$ is the inverse of the CDF of $\mathbb{P}$.*

Recall that Problem 2 prescribes a bandwidth $\lambda$, and requires finding a uniqueness set for graphon signals with the prescribed bandwidth. Let $K$ be the number of eigenvalues of $\mathbf{W}$ which are smaller than $\lambda$ (i.e., $K = \sup\{k \mid \lambda_k < \lambda\}$). The following result shows that $K$ is precisely the number of elements or samples that we need to add to the graphon uniqueness set.

**Proposition 2.** *There exists a set of functions $\{f_i\}_{i=1}^K$, called frames, such that for any graphon signal $X \in \mathrm{PW}_\lambda(\mathbf{W})$ there is a unique reconstruction of $X$ from samples $\{\langle f_i, X \rangle\}_{i=1}^K$.*

To see why this result is possible, recall that if $X \in \mathrm{PW}_\lambda(\mathbf{W})$ for some $\lambda < \lambda_{K+1}$ then $X$ is a linear combination of $K$ eigenfunctions $\{\varphi_i\}_{i=1}^K$ corresponding to $\{\lambda_i\}_{i=1}^K$. Therefore, it suffices to calculate $K$ coefficients $\mathbf{c} = (c_i)_{i=1}^K$ by forming a full rank system (if one exists), which can then be solved via Gaussian elimination:

$$\begin{pmatrix} \langle f_1, \varphi_1 \rangle & \langle f_1, \varphi_2 \rangle & \cdots & \langle f_1, \varphi_K \rangle \\ \langle f_2, \varphi_1 \rangle & \langle f_2, \varphi_2 \rangle & \cdots & \langle f_2, \varphi_K \rangle \\ \vdots & \vdots & \cdots & \vdots \\ \langle f_K, \varphi_1 \rangle & \langle f_K, \varphi_2 \rangle & \cdots & \langle f_K, \varphi_K \rangle \end{pmatrix} \mathbf{c} = \begin{pmatrix} \langle f_1, X \rangle \\ \langle f_2, X \rangle \\ \vdots \\ \langle f_K, X \rangle \end{pmatrix}$$

The next result tells us that different choices of mixture components and $\mathbf{k}$ result in frames with different approximation quality. Specifically, the approximation quality is a function of how well-separated the components in $\mathbb{P}$ are with respect to $\mathbf{k}$ and is measured quantitatively by a difficulty function $\phi(\mathbb{P}, K)$ (Schiebinger et al., 2015). E.g., if there are repeated components in the mixture, or a bimodal component, we expect $\phi$ to be high.

**Proposition 3.** *When $\mathbf{W}$ is viewed as a mixture model of random graphs $\mathbb{K}(\Omega, \mathbb{P}, \mathbf{k})$ with $K$ components $\{\mathbb{P}_i\}_{i=1}^K$ the square-root kernelized density $\{q_i := \sqrt{\int_\Omega \mathbf{k}(\Omega, \cdot)\mathrm{d}\mathbb{P}_i(\Omega)}\}_{i=1}^K$ is a good frame approximation. Quantitatively, let $\mathbf{\Phi}$ be the subspace spanned by the eigenfunctions of $\mathbf{W}$ corresponding to $\{\lambda_i\}_{i=1}^K$, and $\mathbf{Q}$ the subspace spanned by the $\{q_i\}_{i=1}^K$. Then:*

$$\|\Pi_{\mathbf{\Phi}} - \Pi_{\mathbf{Q}}\|_{HS} \leq 16\sqrt{12 + b}\phi(\mathbb{P}, \mathbf{k}), \tag{6}$$

*where $\|.\|_{HS}$ is the Hilbert-Schmidt norm, $\Pi$ is the projection operator, and the difficulty function $\phi$ and the boundedness parameter $b$ are as in (Schiebinger et al., 2015) and App. F[1].*

Next, we connect the square-root kernelized density $q_i$ back to graphon uniqueness sets. The following result shows that when the $q_i$'s align with eigenfunctions of $\mathbf{W}$, there is a clear correspondence between the uniqueness set and the mixture components. The proof is in App. D.

**Theorem 4.** *Fix a small $\epsilon > 0$. Assuming that $\|q_i - \varphi_i\|_{L^2(\mathbb{P}_i)} < \epsilon$ for all $i \in [K]$; and that there exists a set of disjoint measurable subsets $\{A_i \subset [0,1]\}_{i=1}^K$ such that EITHER:*

- *the kernelized density $p_i := \int_{\mathcal{X}} \mathbf{k}(\omega, \cdot)\mathrm{d}\mathbb{P}_i(\omega)$ is concentrated around an interval $A_i \subset [0,1]$ in the sense that $p_i(A_i) - K^2\epsilon^2 > \sum_{i' \neq i} p_i(A_{i'})/(K-1)^2$ for each $i \in [K]$, OR*

- *for each $i \in [K]$, the likelihood ratio statistic is large: $\frac{p_i(A_i) - K^2\epsilon^2}{\sum_{k \neq i} p_k(A_i)} > 1/(K-1)^2$,*

*then the set $U = \bigcup_{i=1}^K A_i$ is a uniqueness set for $\mathrm{PW}_\lambda(\mathbf{W})$ for any $\lambda \in (\lambda_K, \lambda_{K+1})$.*

Put together, the above results culminate in a method to find uniqueness sets by recovering the mixture components. However, this is still cumbersome to implement due to the continuous nature of graphons. Next we explore an efficient approach to find approximate uniqueness sets for a graphon by finding uniqueness sets for a finite graph sampled from (and thus converging to[2]) the graphon.

**Gaussian elimination (GE) on (approximations) of graphon eigenfunctions returns uniqueness sets for finite sampled graphs.** We now derive a scheme to sample points $\omega$ from a uniqueness set $U$ with high probability. Assume that from $\mathbb{W} = \mathbb{K}$, we sample $n$ points to collect a dataset $\{\omega_i\}_{i=1}^n$. From a graphon perspective, these points are nodes in a finite graph $\mathbf{G}_n$ of size $n$ where the edges are sampled with probability given by $\mathbf{W}$. From a mixture model perspective, the points $\omega_i \in \Omega$ are associated with a latent variable $\{z_i \in [K]\}_{i=1}^n$ that indicates the component the sample came from. By building on a result by Schiebinger et al. (2015) on the geometry of spectral clustering, we can unify these two perspectives: running a variant of GE over the Laplacian eigenvectors of a large enough $\mathbf{G}_n$ returns a sample from each mixture component with high probability.

**Theorem 5.** *For any $t > c_0\sqrt{\phi_n(\delta)}w_{min}^{-3}$, GE over the Laplacian eigenvectors of $\mathbf{G}_n$ recovers $K$ samples distributed according to each of the mixture components $\mathbb{P}_i$, $1 \leq i \leq K$, with probability at least*

$$\left(1 - 8K^2\exp{-\frac{c_2 n\delta^4}{\delta^2 + S_{max} + C}}\right)\frac{(1-\alpha)^K(N - n_{min})^K}{(N - (1+\alpha)n_{min})^K}, \text{ with } n_{\min} = \min_{m \in [K]}|\{i : z_i = m\}|, \tag{7}$$

*where $\alpha$ is upper bounded as $\alpha \leq c_1\phi_n(\delta)/w_{min}^{3/2} + \psi(2t)$. The constants $c_1$, $c_2$, $w_{min}$ and $\delta$, and the functions $C$, $S$, $\phi_n$ and $\psi$ are as in (Schiebinger et al., 2015) and App. F.*

Prop. 5 in App. D works out a small example, corresponding to a case where the $\mathbb{P}_i$'s are uniformly distributed on disjoint domains. There, we show that by using GE, we end up solving an eigenvector problem of order $K$, the number of components, instead of the naive order $n \gg K$.

Intuitively, for well-separated mixture models, embedding the dataset via the top Laplacian eigenvectors returns an embedded dataset that exhibits an almost orthogonal structure: points that share the same latent variable (i.e., which came from the same mixture component) have a high probability of lying along the same axis in the orthogonal system; while points sampled from different distributions tend to be positioned orthogonally. GE with proper pivoting on $\mathbf{G}_n$ is thus a good heuristic for sampling uniqueness sets, as it selects points that are almost orthogonal to each other, which is

---

[1]For completeness, we have define and discuss the parameters of the difficulty function in App. F.

[2]Sequences of graphs sampled from a graphon are always convergent (Borgs et al., 2008).

equivalent to picking a sample from each component. The significance of this result is twofold: it bridges graphon sampling and kernelized spectral clustering; and the almost orthogonal structure ensures that the set sampled via GE is a uniqueness set for large graphs sampled from $\mathbf{W}$ with high probability. This is stated in the following proposition, which we prove in App. D.

**Proposition 4.** *Consider a graph sequence* $\mathbf{G}_n \xrightarrow{n \to \infty} \mathbf{W}_3$ *If there is a* $\delta \in (0, \|\mathbf{k}\|_{\mathbb{P}}/(b\sqrt{2\pi}))$ *such that the difficulty function[3] is small, i.e.,* $\phi_n(\delta) < \left(\frac{w_{\min}^3 t}{(3/\pi+1)c_0}\right)^2$, *then with probability at least that in Thm. 5, there exists a minimum number of nodes* $N$ *such that, for all* $n > N$, *the sampled nodes form a uniqueness set for the finite graph* $\mathbf{G}_n$. *All quantities in the bound and additional assumptions are the same as in (Schiebinger et al., 2015) and App. F.*

## 5 ALGORITHM

Motivated by Theorems 3–5, we propose a novel algorithm for efficient sampling of signals on large graphs via graphon signal sampling. When the regularity assumptions of our theorems are satisfied, this algorithm will generate a consistent sampling set.

Consider a graph $\mathbf{G}_n = (\mathcal{V}, \mathcal{E})$ and signal $\mathbf{x}_n$ from which we want to sample a subgraph $\mathbf{G}_m$ and signal $\mathbf{x}_m$ with minimal loss of information (i.e., we would like the signal $\mathbf{x}_n$ to be uniquely represented on the sampled graph $\mathbf{G}_m$). The proposed algorithm consists of three steps:

(1) Represent $\mathbf{G}_n$ as its induced graphon $\mathbf{W}_n(\omega, \theta) = \sum_{i=1}^n \sum_{j=1}^n [\mathbf{A}_n]_{ij} \mathbb{I}(\omega \in I_i) \mathbb{I}(\theta \in I_j)$ where $I_1 \cup \ldots \cup I_n$ is the $n$-equipartition of $[0, 1]$.

(2) Define a coarser equipartition $I'_1 \cup \ldots \cup I'_q$, $q < n$, of $[0, 1]$. Given the bandwith $\lambda$ of the signal $\mathbf{x}_n$, sample a graphon uniqueness interval $\cup_{j=1}^p I'_{i_j}$ (Def. 2), $p < q$, from $I'_1 \cup \ldots \cup I'_q$.

(3) Sample the graph $\mathbf{G}_m$ by sampling $r = \lfloor m/(p-1) \rfloor$ points from each of the $I'_{i_1}, \ldots, I'_{i_{p-1}}$ in the graphon uniqueness set (and the remaining $m - (p-1)r$ nodes from $I_{i_p}$). By Prop. 4, this procedure yields a uniqueness set for $\mathbf{G}_n$ with high probability.

To realize (2), we develop a heuristic based on representing the graphon $\mathbf{W}_n$ on the partition $I'_1 \cup \ldots \cup I'_q$ as a graph $\tilde{\mathbf{G}}_q$ with adjacency matrix given by $[\tilde{\mathbf{A}}_q]_{ij} = \int_{I'_i} \int_{I'_j} \mathbf{W}_n(x, y) \mathrm{d}x \mathrm{d}y$. We then sample $p$ nodes from $\tilde{\mathbf{G}}_q$—each corresponding to an interval $I'_{i_j} \subset I'_1 \cup \ldots \cup I'_q$—using the graph signal sampling algorithm from (Anis et al., 2016). This algorithm is a greedy heuristic closely connected to GE and E-optimal sampling but without spectral computations.

The sampling of $m$ nodes from $I'_{i_1} \cup \ldots \cup I'_{i_p}$ in step (3) is flexible in the way nodes in each interval are sampled. Random sampling is possible, but one could design more elaborate schemes based on local node information. To increase node diversity, we employ a scheme using a local clustering algorithm based on the localized heat kernel PageRank (Chung & Simpson, 2018) to cluster the graph nodes into communities, and then sample an equal number of nodes from each community.

**Runtime analysis.** The advantages of algorithm (1)–(3) w.r.t. conventional graph signal sampling algorithms (e.g., (Anis et al., 2016; Marques et al., 2015)) are twofold. First, if $q \ll n$, (2) is much cheaper. E.g., the heuristic from (Anis et al., 2016) now costs $O(pq^2)$ as opposed to $O(p|\mathcal{E}|)$. If step (3) uses uniform sampling then our method runs in $O(|\mathcal{E}| + pq^2 + m)$; whereas obtaining a uniqueness set of size $m$ from Anis et al. (2016) requires $O(m|\mathcal{E}|)$ time. Second, given the graphon $\mathbf{W}$, we only need to calculate the sampled intervals once and reuse them to find approximate uniqueness sets for any graph $\mathbf{G}_n$ generated from $\mathbf{W}$ as described in Section 2.2, provided that their node labels $\omega_1, \ldots, \omega_n$ (or at least their order) are known. Thus we save time on future sampling computations.

## 6 NUMERICAL EXPERIMENTS

**Transferability for node classification.** We use our sampling algorithm to subsample smaller graphs for training GNNs that are later transferred for inference on the full graph. We consider node classification on citation networks (Yang et al., 2016) and compare the accuracy of GNNs trained on the full graph, on graphs subsampled following the proposed algorithm, and on graphs

---

[3]Notice a slight reparameterization.

Table 1: Accuracy and runtime for models trained on the full graph, a graphon-subsampled graph, and a subgraph with randomly sampled nodes with the same size as (ii). The columns correspond to doubling the number of communities, doubling $r$, and doubling the eigenvalue index.

| | Cora | | | | CiteSeer | | | |
|---|---|---|---|---|---|---|---|---|
| | base | x2 comm. | x2 nodes per int. | x2 eig. | base | x2 comm. | x2 nodes per int. | x2 eig. |
| full graph | $0.86 \pm 0.02$ | $0.86 \pm 0.01$ | $0.86 \pm 0.01$ | $0.85 \pm 0.01$ | $0.80 \pm 0.01$ | $0.81 \pm 0.01$ | $0.79 \pm 0.01$ | $0.79 \pm 0.02$ |
| graphon sampl. | $\mathbf{0.49 \pm 0.09}$ | $\mathbf{0.56 \pm 0.09}$ | $\mathbf{0.73 \pm 0.05}$ | $0.51 \pm 0.09$ | $\mathbf{0.56 \pm 0.06}$ | $\mathbf{0.56 \pm 0.05}$ | $\mathbf{0.67 \pm 0.03}$ | $0.51 \pm 0.10$ |
| random sampl. | $0.46 \pm 0.09$ | $0.52 \pm 0.17$ | $0.71 \pm 0.05$ | $\mathbf{0.57 \pm 0.14}$ | $0.51 \pm 0.08$ | $0.48 \pm 0.11$ | $\mathbf{0.67 \pm 0.03}$ | $\mathbf{0.52 \pm 0.03}$ |

| | PubMed | | | | runtime (s) | | | |
|---|---|---|---|---|---|---|---|---|
| | base | x2 comm. | x2 nodes per int. | x2 eig. | Cora | CiteSeer | PubMed | |
| full graph | $0.76 \pm 0.02$ | $0.77 \pm 0.02$ | $0.77 \pm 0.03$ | $0.77 \pm 0.01$ | 0.9178 | 0.8336 | 0.8894 | |
| graphon sampl. | $\mathbf{0.71 \pm 0.07}$ | $0.67 \pm 0.06$ | $\mathbf{0.75 \pm 0.05}$ | $0.69 \pm 0.07$ | $\mathbf{0.3091}$ | 0.2578 | $\mathbf{0.3204}$ | |
| random sampl. | $0.69 \pm 0.07$ | $\mathbf{0.71 \pm 0.07}$ | $0.74 \pm 0.07$ | $\mathbf{0.72 \pm 0.04}$ | 0.3131 | $\mathbf{0.2514}$ | 0.3223 | |

Table 2: Accuracy and PE compute runtime on MalNet-Tiny w/o PEs, w/ PEs computed on full graph, w/ PEs computed on graphon-sampled subgraph (removing or not isolated nodes), and w/ PEs computed on subgraph with randomly sampled nodes (removing or not isolated nodes).

| | no PEs | full graph PEs | graphon sampl. PEs | | randomly sampl. PEs | | PE compute runtime (s) | |
|---|---|---|---|---|---|---|---|---|
| | | | w/ isolated | w/o | w/ isolated | w/o | | |
| mean | $0.26\pm0.03$ | $0.43\pm0.07$ | $\mathbf{0.29\pm0.06}$ | $\mathbf{0.33\pm0.06}$ | $0.28\pm0.07$ | $0.27\pm0.07$ | full | 12.40 |
| max | 0.30 | 0.51 | $\mathbf{0.40}$ | $\mathbf{0.42}$ | 0.35 | 0.37 | sampl. | 0.075 |

sampled at random. To ablate the effect of different parameters, we consider a base scenario and 3 variations. For Cora and CiteSeer, the base scenario fixes the cutoff frequency at the 5th smallest eigenvalue, $\lambda_5$, of the full graph. It partitions $[0, 1]$ into $q = 20$ intervals and samples $p = 10$ intervals from this partition in step (2). In step (3), it clusters the nodes in each sampled interval into 2 communities and samples $r = 20$ nodes from each sampled interval, 10 per community. For PubMed, the parameters are the same except $q = 30$ and $p = 15$. The three variations are doubling (i) the number of communities, (ii) $r$, and (iii) the eigenvalue index. Further details are in App. G.

Table 1 reports results for 5 realizations. Graphon sampling performs better than random sampling in the base case, where the subsampled graphs have less than 10% of the full graph size. Increasing the number of communities improves performance for Cora and widens the gap between graphon and random sampling for both Cora and CiteSeer. For PubMed, it tips the scale in favor of random sampling, which is not very surprising since PubMed has less classes. When we double $r$, the difference between graphon and random sampling shrinks as expected. Finally, when we increase $\lambda$, graphon sampling performs worse than random sampling. This could be caused by the sample size being too small to preserve the bandwith, thus worsening the quality of the sampling sets.

**Positional encodings for graph classification.** Many graph positional encodings (PEs) for GNNs and graph transformers use the first $K$ normalized Laplacian eigenvectors (or their learned representations) as input signals (Dwivedi et al., 2021; Lim et al., 2022); they provide additional localization information for each node. While they can greatly improve performance, they are expensive to compute for large graphs. In this experiment, we show how our algorithm can mitigate this issue. We sample subgraphs for which the Laplacian eigenvectors are computed, and then use these eigenvectors as PEs for the full-sized graph by zero-padding them at the non-sampled nodes.

We consider the MalNet-Tiny dataset (Freitas et al., 2021), modified to anonymize the node features and pruned to only keep large graphs (with at least 4500 nodes). After balancing the classes, we obtain a dataset with 216 graphs and 4 classes on which we compare four models: (i) without PEs, and qith PEs calculated from (ii) the full-sized graph, (iii) a graphon-sampled subgraph, and (iv) a randomly sampled subgraph. For (iii) and (iv), we also consider the case where isolated nodes are removed from the sampled graphs to obtain more meaningful PEs.

We report results for 10 random realizations in Table 2. The PEs from the graphon-subsampled graphs were not as effective as the PEs from the full-sized graph, but still improved performance with respect to the model without PEs, especially without isolated nodes. In contrast, on average, PEs from subgraphs with randomly sampled nodes did not yield as significant an improvement, and displayed only slightly better accuracy than random guessing when isolated nodes were removed.

## ACKNOWLEDGEMENTS

TL and SJ were supported by NSF awards 2134108 and CCF-2112665 (TILOS AI Institute), and Office of Naval Research grant N00014-20-1-2023 (MURI ML-SCOPE). This work was done while LR was at MIT, supported by a METEOR and a FODSI fellowships.

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

## A    EXTRA NOTATIONS

For some probability measure $\mathbb{Q}$, and some functions in the same $L^2(\mathbb{Q})$ spaces, denote by $\langle \cdot, \cdot \rangle_{L^2(\mathbb{Q})}$ the $L^2(\mathbb{Q})$ inner product and $\| \cdot \|_{L^2(\mathbb{Q})}$ the induced $L^2$ norm. We will also abuse notation and write $L^2(D)$ for some set $D$ that is a closed subset of the real line to mean the $L^2$ space supported on $D$ under the usual Lebesgue measure. When the measure space is clear, we will also drop it and simply write $L^2$.

For some set of functions $\{f_1, \ldots f_K\}$, $\{g_1, \ldots, g_K\}$ where $f_i$ and $g_j$ are in the same $L^2$ space, denote by $((f_i, g_j))_{i,j=1}^K$ the $K \times K$ matrix:

$$((f_i, g_j))_{i,j=1}^K = \begin{bmatrix} \langle f_1, g_1 \rangle_{L^2} & \langle f_1, g_2 \rangle_{L^2} & \cdots & \langle f_1, g_K \rangle_{L^2} \\ \langle f_2, g_1 \rangle_{L^2} & \langle f_2, g_2 \rangle_{L^2} & \cdots & \langle f_2, g_K \rangle_{L^2} \\ \vdots & \vdots & \cdots & \vdots \\ \langle f_K, g_1 \rangle_{L^2} & \langle f_K, g_2 \rangle_{L^2} & \cdots & \langle f_K, g_K \rangle_{L^2} \end{bmatrix} \tag{8}$$

## B    ADDITIONAL BACKGROUND

In this section, we revisit operator theory arguments in our construction of various graphon objects (degree function, normalized graphon, graphon shift operators and normalized graphon Laplacian) from Section 2.2.

Recall that a graphon $\mathbf{W}$ is a bounded, symmetric and $L^2$-measurable function from $[0,1]^2 \to [0,1]$ and thus induces a Hilbert-Schmidt kernel with open connected domain $\mathbf{W} : (0,1)^2 \to [0,1]$. We will abuse notation and refer to both of these objects as graphons. The associated Hilbert-Schmidt integral operator for $\mathbf{W}$ is:

$$H : L^2([0,1]) \to L^2([0,1]) : X \mapsto \left( v \mapsto \int_0^1 \mathbf{W}(u,v)X(u)\mathrm{d}u \right), \tag{9}$$

where the resulting function is understood to be in $L^2$. When $\mathbf{W}$ is viewed as the adjacency matrix of a graph with infinitely many vertices, if $X$ is taken to assign each nodes with a feature in $[0,1]$, then $H$ is understood as a message-passing operator that aggregates neighboring features into each node. Note that measurable functions are only defined up to a set of measure 0.

In the paper, we consider a normalized version of $\mathbf{W}$:

$$\overline{\mathbf{W}}(u,v) = \begin{cases} \mathbf{W}(u,v)/\sqrt{\mathbf{d}(u)\mathbf{d}(v)} & \text{if } \mathbf{d}(u) \neq 0 \text{ and } \mathbf{d}(v) \neq 0 \\ 0 & \text{otherwise.} \end{cases} \tag{10}$$

where $\mathbf{d} \in L^2([0,1])$ is the degree function:

$$\mathbf{d}(u) = \int_0^1 \mathbf{W}(u,v)\mathrm{d}v. \tag{11}$$

It is clear that $\overline{\mathbf{W}}$ is also bounded, symmetric and $L^2$-measurable. The corresponding HS operator is denotes $\overline{H}$. When the kernel is symmetric and has bounded $L^2([0,1]^2)$-norm, then Hilbert-Scmidt operator theory tells us that $H$ is continuous, compact and self-adjoint.

Spectral theory of HS operators then tell us that $H$ and $\overline{H}$ has countable discrete spectrum $\{\lambda_1 \geq \lambda_2 \geq \ldots\}, \{\overline{\lambda}_1 \geq \overline{\lambda}_2 \geq \ldots\}$ and the essential spectrum of a single accumulation point 0 (Lovász, 2012). Furthermore, each nonzero eigenvalues have finite multiplicity (Lovász, 2012). As compact self-adjoint operator, $H$ and $\overline{H}$ admits a spectral theorem:

$$\mathbf{W}(u,v) \sim \sum_{k \in \mathbb{N}} \lambda_k \varphi_k(u)\varphi_k(v), \tag{12}$$

for some eigenfunctions $\{\varphi_k\}_{k \in \mathbb{N}}, \|\varphi_k\|_{L^2} = 1$ (Lovász, 2012).

Recall that Mercer's theorem asserts that continuous positive semi-definite kernel $k$ admits a spectral theorem: there exists a set of orthonormal functions $\{p_i\}_{i \in \mathbb{N}}$ and a countable set of eigenvalues $\{\lambda_i\}_{i \in \mathbb{N}}$ such that $\sum_{i=1}^{\infty} \lambda_i p_i(u)p_j(v) = k(u,v)$ where the convergence is absolute and uniform. For measurable kernels (graphons), Eq. (12) only converges in $L^2$ norm. However, the sequence of eigenvalues admits a stronger $\ell^2$ convergence:

$$\sum_{i=1}^{\infty} \lambda_i^2 = \|\mathbf{W}\|_2^2. \tag{13}$$

Note that by our normalization, $\|\overline{\mathbf{W}}\|_2^2 \leq 1$ and thus $|\overline{\lambda}_i| \leq 1$ for all $i \in \mathbb{N}$. Finally, we defined the normalized Laplacian operator $\overline{\mathcal{L}} = \mathrm{Id} - \overline{H}$. It is then straightforward to see that the spectrum of $\overline{\mathcal{L}}$ is just $1 - \sigma(\overline{H})$ set-wise.

## C  POINCARÉ INEQUALITY

*Proof of Thm. 2.* The proof mirrors Pesenson (2008). Fix an $X$ in $L^2(U)$. Define $X' \in L^2(D)$ as:

$$X'(u) = \begin{cases} X(u) & \text{if } u \in U \\ -X(u) & \text{if } u \in U' \\ 0 & \text{otherwise.} \end{cases} \tag{14}$$

It is clear that $X'$ is measureable (with respect to Lebesgue measure on $D$). Consider:

$$\|X'(u)\|_{L^2(D)}^2 = \int_U (X'(u))^2 \mathrm{d}u + \int_{U'} (X'(u))^2 \mathrm{d}u = 2\|X(u)\|_{L^2(U)}^2, \tag{15}$$

and at the same time, for all $u \in U$:

$$\int_0^1 \mathbf{W}(u,v)\mathrm{d}v = \int_{U \cup \mathcal{N}(U)} \mathbf{W}(u,v)\mathrm{d}v = \int_D (\Gamma(U))(u,v)\mathrm{d}v. \tag{16}$$

This in particular means that normalizing $\Gamma(U)$ as $\Gamma(U)'$ means scaling by the same scalar as normalizing $\mathbf{W}$ into $\mathbf{W}'$.

Now we investigate the image of $X'$ under Laplacian operator:

$$L'_{\Gamma(U)}X'(u) := X'(u) - \int_D (\Gamma(U))'(u,v)X'(v)\mathrm{d}v \tag{17}$$

$$= \begin{cases} X(u) - \int_0^1 \mathbf{W}'(u,v)X(v)\mathrm{d}v & \text{if } u \in U \\ -X(u) - \int_0^1 -\mathbf{W}'(u,v)X(v)\mathrm{d}v & \text{if } u \in U' \\ 0 & \text{otherwise,} \end{cases} \tag{18}$$

$$= \begin{cases} \mathcal{L}'X(u) & \text{if } u \in U \\ -\mathcal{L}'X(u) & \text{if } u \in U' \\ 0 & \text{otherwise.} \end{cases} \tag{19}$$

And therefore: $\|\mathcal{L}'_{\Gamma(U)}X'\|_{L^2(D)} = \sqrt{2}\|\mathcal{L}'X\|_{L^2(U)} \leq \sqrt{2}\|\mathcal{L}'X\|_{L^2([0,1])}$. The point of constructing $\Gamma(U)'$ is that it has a nice eigenfunction that corresponds to eigenvalue 0. Let $\varphi_0$ be such a function, then

$$0 = \mathcal{L}'_{\Gamma(U)}\varphi_0(u) = \varphi_0(u) - \int_D \frac{(\Gamma(U))(u,v)}{\sqrt{\int_D(\Gamma(U))(z,v)\mathrm{d}z \int_D(\Gamma(U))(u,z)\mathrm{d}z}}\varphi_0(v)\mathrm{d}v. \tag{20}$$

By inspection, setting $\varphi_0(u) := \sqrt{\int_D(\Gamma(U))(u,v)\mathrm{d}v}$ satisfies the above equation and this is the eigenfunction of $\mathcal{L}'_{\Gamma(U)}$ corresponding to eigenvalue 0. Expand $X'$ in the eigenfunction basis of $\mathcal{L}'_{\Gamma(U)}$ to get:

$$\|X'\|_{L^2(D)} = \sum_{i \in \mathbb{N} \cup \{0\}} |\langle X', \varphi_i \rangle|^2. \tag{21}$$

However, the first coefficient vanishes:

$$\langle X', \varphi_0 \rangle = \int_D X'(u)\sqrt{\int_D(\Gamma(U))(u,v)\mathrm{d}v}\mathrm{d}u \tag{22}$$

$$= \int_U X(u)\sqrt{\int_D \mathbf{W}(u,v)\mathrm{d}v}\mathrm{d}u - \int_{U'} X(u)\sqrt{\int_D \mathbf{W}(u,v)\mathrm{d}v}\mathrm{d}u = 0, \tag{23}$$

and we have:

$$\sqrt{2}\|\mathcal{L}'X\|_{L^2([0,1])} \geq \|\mathcal{L}'_{\Gamma(U)}X'\|^2_{L^2(D)} \tag{24}$$

$$= \sum_{i \in \mathbb{N}} \lambda_i^2 |\langle f', \varphi_i \rangle|^2 \tag{25}$$

$$\geq \lambda_1^2 \|X'\|^2_{L^2(D)} \tag{26}$$

$$= \sqrt{2}\|X\|^2_{L^2(U)}, \tag{27}$$

which finishes the proof. $\qquad\square$

*Proof of Thm. 3.* If $X, Y \in PW_\lambda(\mathbf{W})$, then $X - Y \in PW_\lambda(\mathbf{W})$ and we have:

$$\|\bar{\mathcal{L}}(X-Y)\|_{L^2} \leq \lambda\|X-Y\|_{L^2}. \tag{28}$$

If $X$ and $Y$ coincide on $U$, then $X - Y \in L^2(S)$ and we can write the Poincaré inequality:

$$\|X-Y\|_{L^2} \leq \Lambda\|\bar{\mathcal{L}}(X-Y)\|_{L^2}. \tag{29}$$

Combining the two inequalities, we have:

$$\|X-Y\|_{L^2} \leq \Lambda\|\bar{\mathcal{L}}(X-Y)\|_{L^2} \leq \Lambda\lambda\|X-Y\|_{L^2} \tag{30}$$

which can only be true if $\|X-Y\|_{L^2} = 0$ since $\lambda\Lambda < 1$. $\qquad\square$

# D  PROOF FROM SECTION 4.2

## D.1  GRAPHON IS EQUIVALENT TO MIXTURE MODEL FOR RANDOM GRAPHS

*Proof of Prop. 1.* Let $\omega \sim \mathbb{P}(\Omega)$. We want to find a strictly monotone function $\beta : [0, 1] \to \Omega$ such that $U = \beta^{-1}(\omega)$ is uniformly distributed over $[0, 1]$. Let $F_{\omega}(\omega) = \mathbb{P}(\omega \leq \omega)$, and assume the function $\beta$ exists. Then, for all $\omega$ we can write

$$F_{\omega}(\omega) = \mathbb{P}(\omega \leq \omega) = \mathbb{P}(\beta(U) \leq \omega) = \mathbb{P}(U \leq \beta^{-1}(\omega)) = \beta^{-1}(\omega) \tag{31}$$

where the second equality follows from the fact that, since $\beta$ is strictly monotone, it has an inverse. This proves that $\beta$ exists and is equal to the inverse of the CDF of $\omega$. $\square$

Before continuing, let us introduce a few useful definitions. The Laplacian associated with the model $\mathbb{K}(\Omega, \mathbb{P}, K)$ is defined as

$$\mathcal{L}_K f = f - \int_{\Omega} \bar{K}(\omega, \cdot) f(\omega) d\mathbb{P}(\omega) \tag{32}$$

where $\bar{K}(\omega, \theta) = K(\omega, \theta)/(q(\omega)q(\theta))$ and $q(\omega) = \sqrt{\int_{\Omega} \bar{K}(\omega, \theta) d\mathbb{P}(\theta)}$. The operator $\mathcal{L}$ is self-adjoint and positive semidefinite, therefore it has a non-negative real spectrum $\{\lambda_i, \varphi_i\}_{i=1}^{\infty}$.

To simplify matters, we will consider the problem of finding frames $\{f_i\}_{i=1}^{K}$ allowing to uniquely represent signals in any $PW_{\Omega}(\lambda)$ with $\lambda \leq \lambda_K$. Note that the graphon $\mathbf{W}$ (and therefore its associated Laplacian) are themselves rank $K$. Recall that, in order to uniquely represent a signal, the frame $\{f_i\}$ must satisfy

$$\mathrm{rank} \begin{bmatrix} \langle f_1, \varphi_1 \rangle & \langle f_1, \varphi_2 \rangle & \cdots & \langle f_1, \varphi_K \rangle \\ \langle f_2, \varphi_1 \rangle & \langle f_2, \varphi_2 \rangle & \cdots & \langle f_2, \varphi_K \rangle \\ \vdots & \vdots & \cdots & \vdots \\ \langle f_K, \varphi_1 \rangle & \langle f_K, \varphi_2 \rangle & \cdots & \langle f_K, \varphi_K \rangle \end{bmatrix} = K \tag{33}$$

where $\{\varphi_i\}_{i=1}^{K}$ are the eigenfunctions associated with strictly positive eigenvalues of $\mathcal{L}_K$, sorted according to their magnitude.

By (Schiebinger et al., 2015, Thm.1), the functions $q_i(\theta) = \int_{\Omega} K(\omega, \theta) d\mathbb{P}_i(\omega)$, $1 \leq i \leq K$, form such a frame.

## D.2  MIXTURE COMPONENT GIVES RISE TO UNIQUENESS SETS

In this section, we Leb to emphasize the Lebesgue measure on $\mathbb{R}$ being used in our integrals.

*Proof of Thm. 4.* Define the Heaviside frame $\{h_i : \mathcal{X} \to \mathbb{R}\}_{i=1}^{K}$ as $h_i(\omega) = \delta_{\in A_i}(\omega)\sqrt{p_i(\omega)/p_i(A_i)}$ where $\delta_E$ is the Dirac delta function for a measurable set $E$, for each $i \in [K]$. It is straightforward to check that $h_i$ is also in $L^2(p_i)$ for each $i \in [K]$. Define the subspace $\mathbf{H} := \mathrm{span}\{h_1, \ldots, h_K\}$ and the Heaviside embedding $\Phi_{\mathbf{H}} : \mathcal{X} \to \mathbb{R}^K$ as $\Phi_{\mathbf{H}}(\omega) = (h_1(\omega), \ldots, h_K(\omega))$.

**Step 1: Show that** $((h_i, q_j))_{i,j=1}^{K}$ **is full-rank.** To show that $((h_i, q_j))_{i,j=1}^{K}$ is full-rank, we compute entries of $((h_i, q_j))_{i,j=1}^{K}$: for any $i, j \in [K]$,

$$\langle h_i, q_j \rangle = \frac{1}{\sqrt{p_i(A_i)}} \int_{A_i} q_j(\omega)\sqrt{p_i(\omega)}d\mathrm{Leb}(\omega) = \frac{1}{\sqrt{p_i(A_i)}} \int_{A_i} \sqrt{p_j(\omega)p_i(\omega)}d\mathrm{Leb}(\omega). \tag{34}$$

For diagonal entries, note that:

$$\langle h_j, q_j \rangle = \frac{1}{\sqrt{p_j(A_j)}} \int_{A_j} p_j(\omega)d\mathrm{Leb}(\omega) = \sqrt{p_j(A_j)}. \tag{35}$$

Fix an $j \in [K]$ and consider:

$$\sum_{i \neq j} |\langle h_i, q_j \rangle| = \sum_{i \neq j} \frac{1}{\sqrt{p_i(A_i)}} \int_{A_i} \sqrt{p_j(\omega) p_i(\omega)} \mathrm{dLeb}(\omega) \tag{36}$$

$$\leq \sum_{i \neq j} \frac{1}{\sqrt{p_i(A_i)}} \sqrt{\int_{A_i} p_j(\omega) \mathrm{dLeb}(\omega)} \sqrt{\int_{A_i} p_i(\omega) \mathrm{dLeb}(\omega)} \tag{37}$$

$$= \sum_{i \neq j} \frac{1}{\sqrt{p_i(A_i)}} \sqrt{p_j(A_i)} \sqrt{p_i(A_i)} \tag{38}$$

$$= \sum_{i \neq j} \sqrt{p_j(A_i)}, \tag{39}$$

where the inequality is from Cauchy-Schwarz. In the first choice of assumption, we have $p_j(A_j) - K^2 \epsilon^2 > \sum_{i \neq j} p_j(A_i)/(K-1)^2$ and thus $\sqrt{p_j(A_j)} - K\epsilon > \sqrt{\sum_{i \neq j} p_i(A_i)}/(K-1) > \sum_{i \neq j} \sqrt{p_i(A_i)}$, due to monotonicity of square root and Cauchy-Schwarz. Thus, we have shown that for every $j \in [K]$, the $j$-th column of $((h_i, q_j))_{i,j=1}^K$ has $j$-th entry larger (in absolute value) than the sum of absolute values of all other entries. Gershgorin circle theorem then tells us that eigenvalues of $((h_i, q_j))_{i,j=1}^K$ lie in at least one disk center at some diagonal value with radius sum of absolute value of remaining column entries. None of the Gershgorin disks contain the origin, and we can conclude that $((h_i, q_j))_{i,j=1}^K$ has no 0 eigenvalue. Therefore, it is full rank.

Now, fix an $i \in [K]$ and consider:

$$\sum_{j \neq i} |\langle h_i, q_j \rangle| = \sum_{j \neq i} \frac{1}{\sqrt{p_i(A_i)}} \int_{A_i} \sqrt{p_j(\omega) p_i(\omega)} \mathrm{dLeb}(\omega) \tag{40}$$

$$\leq \sum_{j \neq i} \frac{1}{\sqrt{p_i(A_i)}} \sqrt{\int_{A_i} p_j(\omega) \mathrm{dLeb}(\omega)} \sqrt{\int_{A_i} p_i(\omega) \mathrm{dLeb}(\omega)} \tag{41}$$

$$= \sum_{j \neq i} \frac{1}{\sqrt{p_i(A_i)}} \sqrt{p_j(A_i)} \sqrt{p_i(A_i)} \tag{42}$$

$$= \sum_{j \neq i} \sqrt{p_j(A_i)} \tag{43}$$

In the second choice of assumption, the same thing happens: $p_i(A_i) - K^2 \epsilon^2 > \sum_{j \neq i} p_j(A_i)/(K-1)^2$ implies that $\sqrt{p_i(A_i)} - K\epsilon > \sum_{j \neq i} \sqrt{p_i(A_i)}$ and once again, the center of any Gershgorin disk (but this time in the rows) are further away from zero than the sum of absolute value of other non-diagonal entries. Therefore, none of the disks contain the origin and $((h_i, q_j))_{i,j=1}^K$ cannot have 0 eigenvalue, thus full-rank. Therefore, either choices of assumption leads to full-rank-ness of the system $((h_i, q_j))_{i,j=1}^K$.

**Step 2. Full-rank implies uniqueness.** By the premise of this result, we have for each $i$,

$$\|q_i - \varphi_i\|_{L^2} < \epsilon. \tag{44}$$

Thus,

$$\langle h_i, \varphi_j \rangle = \langle h_i, q_j \rangle - \langle h_j, q_j - \varphi_j \rangle \in (\langle h_i, q_j \rangle - \epsilon, \langle h_i, q_j \rangle + \epsilon), \tag{45}$$

by Cauchy-Schwarz.

Recall that $((h_i, q_j))_{i,j}$ is full rank, and that Gershgorin circle theorem applied in the previous step still has a slack of at least $K\epsilon$. Therefore, perturbation element-wise of additive size $\epsilon$ of $((h_i, q_j))_{i,j}$ will still be full rank by Gershgorin circle theorem and we conclude that $((h_i, \varphi_j))_{i,j}$ is full-rank.

Let $X \in \mathrm{PW}_\lambda(\mathbf{W})$ for some $\lambda \in (\lambda_K, \lambda_{K+1})$, then by definition, there exists a vector $\boldsymbol{c} \in \mathbb{R}^K$ such that $X = \sum_{j=1}^K \boldsymbol{c}_j \varphi_j$. Take inner product (in $L^2(\mathbb{P}) = L^2(\mathbf{W})$), we have:

$$
\begin{bmatrix}
\langle h_1, \varphi_1 \rangle & \langle h_1, \varphi_2 \rangle & \dots & \langle h_1, \varphi_K \rangle \\
\langle h_2, \varphi_1 \rangle & \langle h_2, \varphi_2 \rangle & \dots & \langle h_2, \varphi_K \rangle \\
\vdots & \vdots & \dots & \vdots \\
\langle h_K, \varphi_1 \rangle & \langle h_K, \varphi_2 \rangle & \dots & \langle h_K, \varphi_K \rangle
\end{bmatrix}
\boldsymbol{c} =
\begin{bmatrix}
\langle h_1, X \rangle \\
\langle h_2, X \rangle \\
\vdots \\
\langle h_K, X \rangle
\end{bmatrix}
\tag{46}
$$

.

To test if $U = \bigcup_{i=1}^K A_i$ is a uniqueness set, we assume that $\|X\delta_U\|_{L^2(\mathbf{W})} = 0$. But $|\langle h_i, X \rangle| = |\langle h_i, \delta_{A_i} X \rangle| \le \|h_i\| \|X\delta_U\| = 0$ for each $i$ in $[K]$. Thus:

$$
\begin{bmatrix}
\langle h_1, \varphi_1 \rangle & \langle h_1, \varphi_2 \rangle & \dots & \langle h_1, \varphi_K \rangle \\
\langle h_2, \varphi_1 \rangle & \langle h_2, \varphi_2 \rangle & \dots & \langle h_2, \varphi_K \rangle \\
\vdots & \vdots & \dots & \vdots \\
\langle h_K, \varphi_1 \rangle & \langle h_K, \varphi_2 \rangle & \dots & \langle h_K, \varphi_K \rangle
\end{bmatrix}
\boldsymbol{c} =
\begin{bmatrix}
0 \\
0 \\
\vdots \\
0
\end{bmatrix}
\tag{47}
$$

.

Finally, since $((h_i, \varphi_j))_{i,j=1}^K$ is full rank, its null space is trivial, implying $\boldsymbol{c} = \boldsymbol{0}$ and thus $X = 0$, which proves uniqueness of $U$. □

### D.3 CONSISTENCY THEOREM

This result is an adaptation of (Schiebinger et al., 2015, Thm. 2), which is reproduced below.

**Theorem 6** (Thm.2, Schiebinger et al. (2015))**.** *There are numbers $c$, $c_0$, $c_1$, $c_2$ depending only on $b$ and $r$ such that for any $\delta \in (0, \frac{\|K\|_\mathbb{P}}{b\sqrt{2\pi}})$ satisfying condition (Schiebinger et al., 2015, 3.17) and any $t > c_0 w_{min}^{-1}\sqrt{\phi_n(\delta)}$, the embedded dataset $\{\boldsymbol{\Phi}_\mathcal{V}(\omega_i), Z_i\}_{i=1}^n$ has $(\alpha, \theta)$ orthogonal cone structure with*

$$
|\cos\theta| \le \frac{c_0\sqrt{\phi_n(\delta)}}{w_{min}^3 t - c_0\sqrt{\phi_n(\delta)}}
\tag{48}
$$

$$
\alpha \le \frac{c_1}{w_{min}^{3/2}}\phi_n(\delta) + \psi(2t)
\tag{49}
$$

*and this event holds with probability at least $1 - 8K^2 \exp-\frac{c_2 n\delta^4}{\delta^2 + S_{max} + C}$.*

Thm. 6 elucidates the conditions under which the spectral embeddings of the nodes $\boldsymbol{\omega}$ form an orthogonal cone structure (see (Schiebinger et al., 2015, Def. 1) for a precise definition). This is helpful for Gaussian elimination, as provided that we pick a pivot inside a cone, the other rows to be picked—which are orthogonal to the pivot—are themselves inside other cones, and therefore likely to belong to a different cluster (i.e., to be distributed according to a different mixture component).

We first recall connections between graphons and mixture models and explain how each objects in the context of Thm. 6 can be understood in graphons terms. In mixture model, we sample dataset $\{\omega_i\}_{i=1}^n$ from the mixture distribution. This is equivalent to sampling nodes under a pushforward in when we sample finite graphs from a graphon. Thus, each data point $\omega_i$ is a 'node' of a finite graph sampled from the graphon. Next, the spectral embedding of datapoints in spectral clustering is equivalent to computing the eigenfunction of graphon Laplacian at that datapoint - embedding it in frequency domain. Therefore, from a graphon perspective, the theorem is asserting that given some underlying structure controlled by the difficulty function, embedding of nodes in finite graph from a fix graphon into frequency domain under GFT has a peculiar structure: an orthogonal cone. While we do not have easy access to graphon eigenfunction, computing an approximation once with a large graph suffices. This is because we can reuse the embedding when new points are sampled into the graph!

*Proof of Thm. 5.* Let us consider what happens when performing Gaussian elimination on the columns of $\boldsymbol{\Phi}_\mathcal{V}(\boldsymbol{\omega})$. When picking the pivot, the probability of picking a "good" point inside a cone, i.e., a point that is both inside a cone and that is distributed according to the mixture component associated with that cone, is $1 - \alpha$. Conditioned on this event, the probability of picking a

second "good" point from another cone is $\frac{(1-\alpha)(n-n_1)}{n-(1-\alpha)n_1}$, where $n_1$ is the number of points distributed according to the pivot's distribution, denoted $\mathbb{P}_1$. More generally, the probability of picking a "good" point at the $i$th step, conditioned on having picked $i-1$ "good" points, is

$$\mathbb{P}(i\text{th point is "good"} \mid 1, \ldots, i-1 \text{ are "good"}) = \frac{(1-\alpha)n_{-i}}{n-(1-\alpha)n_{+i}} \tag{50}$$

where $n_{-i} = \sum_{j=1}^{i-1} n - n_j$ and $n_{+i} = n - n_{-i}$.

Since Eq. (50) is a decreasing function of $n_{-i}$, the probability of picking $K$ good points is lower bounded by

$$\mathbb{P}(1, \ldots, K \text{ are "good"}) \geq \frac{(1-\alpha)^K(n-n_{\min})^K}{(n-(1-\alpha)n_{\min})^K} \tag{51}$$

where $n_{\min} = \min_{1 \leq j \leq K} n_j$. Combining Eq. (51) with Theorem Thm. 6 gives the proposition's result. $\qquad\square$

*Proof of Prop. 4.* The conditions on the difficulty function in the hypothesis of Prop. 4 means that the angle $\theta$ in the cone structure is at least $\pi/3$.

Note that every finite graph $\mathbf{G}_n$ induces a graphon via stochastic block model:

$$\mathbf{W}_{\mathbf{G}_n} := \sum_{i=1}^{n}\sum_{j=1}^{n}[\mathbf{A}_n]_{i,j}\mathbb{I}(x \in I_i)\mathbb{I}(y \in I_j) \tag{52}$$

From Ruiz et al. (2021), we know that the eigenvalues of the adjacency HS operator of $\mathbf{W}_{\mathbf{G}_n}$ converges to that of $\mathbf{W}$. As the graphon Laplacian is a scaled and translated operator from the adjacency operator, eigenvalues of the Laplacian also converges. Let the eigenvalues of the finite graph be $\hat{\lambda}_{n,1} \leq \ldots \leq \hat{\lambda}_{n,-1}$ Pick an $n_0$ large enough such that there is a spectral gap $\hat{\lambda}_{n,K} < \hat{\lambda}_{n,K+1}$ for all $n > n_0$. Then pick an even larger $n_1$ such that $\lambda \in (\hat{\lambda}_{n,K}, \hat{\lambda}_{n,K+1})$ for all $n > n_1$. Such a choice of $n_0, n_1$ is guaranteed by convergence of eigenvalue.

Not only do eigenvalue converges, when there is an eigengap, the subspace spanned by the first $K$ eigenfunctions also converges. The convergence is in term of convergence in operator norm of the projection operator (Ruiz et al., 2021). Let the projection operator be $\Phi_{\mathbf{G}_n}$ and $\Phi_{\mathbf{W}}$, corresponding to that for the finite graph $\mathbf{G}_n$ and for the graphon $\mathbf{W}$ respectively. Therefore, we select yet a larger $n_2$ such that $\|\Phi_{\mathbf{G}_n} - \Phi_{\mathbf{W}}\|_{HS} < \epsilon$ for all $n > n_2$ and for some $\epsilon$ to be chosen later.

Recall that we picked some sample via Thm. 5 and with high probability, our sample attains an orthogonal cone structure. In other words, there is a permutation of samples such that for each $i \in [K]$, $|\cos\tau(i)| > 1/2$ with high probability, where $\tau(i)$ is the angle between $\varphi(x_i)$ and the unit vector with all zero entries but the $i$-th one. This means that for any $i$, $|\varphi_i(x_i)|/\|(\varphi_j(x_i))_{j\in[K]}\|_2 > 1/2$. Therefore, the matrix:

$$\begin{bmatrix} \varphi_1(x_1) & \varphi_2(x_1) & \ldots & \varphi_K(x_1) \\ \varphi_1(x_2) & \varphi_2(x_2) & \ldots & \varphi_K(x_2) \\ \vdots & \vdots & \ldots & \vdots \\ \varphi_1(x_K) & \varphi_2(x_K) & \ldots & \varphi_K(x_K) \end{bmatrix} \tag{53}$$

is full rank, since the off-diagonal absolute value sum does not exceed the absolute value of the diagonal entry for every row, via Gershgorin circle theorem. As a corollary from Thm. 1 of Anis et al. (2016), the system being full rank means that the samples drawn form a uniqueness set and the proof is complete.

To select $\epsilon$, notice that there are still slack in Gershgorin circle theorem and one can select such an $\epsilon$ that the two projection has eigenfunctions differs by at most that slack amount in $L^2$. This is possible since full-ranked-ness is a robust property: if a matrix is full-rank then other matrices within a small ball from it is also full-rank. Thus, if there is a converging sequence of eigenfunction/eigenspace to $\varphi(x)$ then the perturbed matrix analogous to Eq. (53) would eventually enter the small ball of full-rank matrices. We leave more precise nonasymptotic analysis to future work.

$\qquad\square$

# E  SMALL EXAMPLE: BLOCK MODEL AND MIXTURE OF DISJOINT UNIFORM DISTRIBUTIONS

Let us consider a simplified setting, consisting of a blockmodel kernel and uniform mixture components, to show an example where Gaussian elimination recovers intervals distributed according to the $\{q_i\}_{i=1}^K$.

**Proposition 5.** *Let $\mathcal{I} = \Omega_1 \cup \ldots \cup \Omega_N$ be an $N$-partition of $\Omega$. Let the kernel $\mathbf{k}$ be a $K$-block model over a coarser partition $\mathcal{I}' = \Omega_1' \cup \ldots \cup \Omega_K'$ of $\Omega$ containing $\mathcal{I}$ (each block has value given by the integral of $K$ over the centroid). Let the $\mathbb{P}_i$ be uniform over the $\Omega_i'$. Then, column-wise Gaussian elimination over the positive eigenfunctions(vectors) finds subsets $\Omega_{j_1}, \ldots, \Omega_{j_K}$ distributed with probability density functions equal to the corresponding $q_i$, up to a normalization.*

*Proof of Prop. 5.* The kernel $\mathbf{k}$ can be written as

$$\mathbf{k}(\omega, \theta) = \sum_{i,j=1}^K a_{ij} \mathbb{I}(\omega \in \Omega_i') \mathbb{I}(\theta \in \Omega_j'). \tag{54}$$

Therefore, the model $\mathbb{K}$ can be represented as a SBM graphon,

$$
\mathbf{A} = 
\begin{array}{c}
\\ i_1^1 \\ \vdots \\ i_{k_1}^1 \\ \\ \vdots \\ \\ i_1^K \\ \vdots \\ i_{k_K}^K
\end{array}
\begin{pmatrix}
\begin{array}{ccccccc}
i_1^1 & \cdots & i_{k_1}^1 & \cdots & i_1^K & \cdots & i_{k_K}^K \\
a_{11} & \cdots & a_{11} & \cdots & a_{1K} & \cdots & a_{1K} \\
\vdots & \ddots & \vdots & & \vdots & \ddots & \vdots \\
a_{11} & \cdots & a_{11} & \cdots & a_{1K} & \cdots & a_{1K} \\
 & & \vdots & & \ddots & & \vdots \\
a_{1K} & \cdots & a_{1K} & \cdots & a_{KK} & \cdots & a_{KK} \\
\vdots & \ddots & \vdots & & \vdots & \ddots & \vdots \\
a_{1K} & \cdots & a_{1K} & \cdots & a_{KK} & \cdots & a_{KK}
\end{array}
\end{pmatrix} \tag{55}
$$

where $i_{j_l}^l$, $1 \leq j_l \leq k_l$, indexes elements of $\mathcal{I}$ contained in $\Omega_l'$ (i.e., in the support of $\mathbb{P}_l$), and $\sum_{j=1}^K k_j = N$. For a more concise representation, let us write

$$\mathbf{A} = \begin{bmatrix} \mathbf{A}_{11} & \cdots & \mathbf{A}_{1K} \\ \vdots & \ddots & \vdots \\ \mathbf{A}_{1K} & \cdots & \mathbf{A}_{KK} \end{bmatrix} \tag{56}$$

where $\mathbf{A}_{ij} = a_{ij} \mathbf{1}\mathbf{1}^T$.

Consider the normalized adjacency $\tilde{\mathbf{A}} = (\mathbf{D}^\dagger)^{1/2} \mathbf{A} (\mathbf{D}^\dagger)^{1/2}$, which has the same block structure as $\mathbf{A}$ but with blocks $\tilde{\mathbf{A}}_{ij}$. Note that technically, we would find eigenvectors of the normalized Laplacian $\mathbf{I} - \tilde{\mathbf{A}}$ but the identity shift only shifts the spectrum by 1 (after inversion about the origin). Therefore it is equivalent to finding the eigenvectors of $\tilde{\mathbf{A}}$:

$$\begin{bmatrix} \tilde{\mathbf{A}}_{11} & \cdots & \tilde{\mathbf{A}}_{1K} \\ \vdots & \ddots & \vdots \\ \tilde{\mathbf{A}}_{1K} & \cdots & \tilde{\mathbf{A}}_{KK} \end{bmatrix} \mathbf{u} = \lambda \mathbf{u}. \tag{57}$$

Note, however, that for each $1 \leq i \leq K$, the rows corresponding to $[\tilde{\mathbf{A}}_{1i} \ldots \tilde{\mathbf{A}}_{Ki}]\mathbf{u}$ are repeated, so plugging $\tilde{\mathbf{A}}$ into an eigensolver without simplifying $\tilde{\mathbf{A}}$ first is going to incur huge computational cost for little gain. We can exploit the repeated structure of $\tilde{\mathbf{A}}$ to do some preprocessing first, via a variant of Gaussian elimination. Permuting the rows and columns of this matrix to ensure the sequence $a_{i1}, \ldots, a_{iK}$ appears in the first $K$ columns, and subtracting the repeated rows, we can

rewrite this as

$$\begin{bmatrix} \tilde{a}_{11} & \dots & \tilde{a}_{1K} & \mathbf{b}_1 \\ \vdots & \ddots & \vdots & \vdots \\ \tilde{a}_{1K} & \dots & \tilde{a}_{KK} & \mathbf{b}_K \\ \mathbf{0} & \dots & \mathbf{0} & \mathbf{0} \end{bmatrix} \mathbf{u} = \lambda \mathbf{u} \tag{58}$$

where the $\mathbf{b}_i \in \mathbb{R}^{N-K}$ are row vectors collecting the remaining entries of row $i$ after permutation, and $\mathbf{0}$ denotes the all-zeros vector of dimension $N - K$.

For the linear system in Eq. (58), it is easy to see that the solutions $\mathbf{u}$ must have form $\mathbf{u} = [u_1 \dots u_k \ 0 \dots 0]^T$. Hence, the eigenvectors of the modified matrix in Eq. (58) are the eigenvectors of its $K \times K$ principal submatrix padded with zeros. To obtain the eigenvectors of the original matrix Eq. (57), we simply have to "revert" the operations performed to get from there to Eq. (58), with the appropriate normalizations to ensure orthonormality. By doing so, we get eigenvectors of the following form

$$\mathbf{u} = \begin{matrix} k_1 \text{ times} \\ \\ \\ \\ \\ k_K \text{ times} \end{matrix} \begin{pmatrix} u_1 \\ \vdots \\ u_1 \\ \\ \vdots \\ \\ u_K \\ \vdots \\ u_K \end{pmatrix} \tag{59}$$

i.e., in every eigenvector of $\tilde{\mathbf{A}}$, entries corresponding to sets $\Omega_i$ contained in the same set $\Omega'_k$ are the same.

Now, assume that we have found all $K$ eigenvectors of $\tilde{\mathbf{A}}$ and collect them in the matrix $\mathbf{U}_K \in \mathbb{R}^{N \times K}$. To find a uniqueness set for the associated graphon, we perform columnwise Gaussian elimination on $\mathbf{U}_K$, and add the indices of the zeros in the $K$th row of the echelon form to the sampling set.

In the current example, this heuristic is always guaranteed to find a uniqueness set. Any combination of indices corresponding to $K$ different rows from $\mathbf{U}_K$ forms such a set. Since through Gaussian elimination we are guaranteed to pick $K$ linearly independent rows, when picking a row from cluster $\Omega_i$ for arbitrary $i$, all $k_i$ rows are equally likely to be picked, as they are equal and thus have the same "pivoting" effect. In an independent trial, the probability of picking a row from $\Omega'_i$ is thus $(k_i/N) \times \mathbb{P}_i$. Up to a normalization, this probability is equal to $\mathbf{q}_i = \mathbf{A}\mathbb{P}_i$. The entries of this vector determine the level sets of $q_i$ as

$$q_i(x) = \mathbf{q}_i \mathbb{I}(x \in \Omega'_i) \tag{60}$$

completing the proof. $\qquad\square$

## F  ELEMENTS FROM (SCHIEBINGER ET AL., 2015)

For completeness, we reproduce elements from (Schiebinger et al., 2015) that were used in our paper.

### F.1  DIFFICULTY FUNCTION FOR MIXTURE MODELS

Recall $\Omega$ is a measurable space and $\mathcal{P}(\Omega)$ is a set of all probability measures on $\Omega$. Let $\mathbb{P}_i \in \mathcal{P}(\Omega)$ mixture components for $i = 1..K$. A mixture model is a convex combination:

$$\mathbb{P} := \sum_{i=1}^{K} w_i \mathbb{P}_i, \tag{61}$$

for a set of weights $w_i \geq 0$ for $i = 1 \dots K$ and $\sum_i w_i = 1$. Recall that there is also a kernel $\mathbf{k}$ associated with the mixture model.

The statistics of how well-separated the mixture components are can be quantified through five defined quantities:

**Similarity index.** For any distinct pair of mixtures $l \neq k$, the kernel-dependent similarity index between $\mathbb{P}_l$ and $\mathbb{P}_k$ is:

$$\mathcal{S}(\mathbb{P}_l, \mathbb{P}_k) := \frac{\int_\Omega \int_\Omega \mathbf{k}(\omega, \theta) \mathrm{d}\mathbb{P}_l(\omega) \mathrm{d}\mathbb{P}_l(\theta)}{\int_\Omega \int_\Omega \mathbf{k}(\omega, \theta) \mathrm{d}\mathbb{P}(\omega) \mathrm{d}\mathbb{P}_l(\theta)}, \tag{62}$$

and the maximum over all ordered pairs of similarity index is:

$$\mathcal{S}_{\max}(\mathbb{P}) := \max_{l \neq k} \mathcal{S}(\mathbb{P}_l, \mathbb{P}_k) \tag{63}$$

In general, $\mathcal{S}_{\max}$ measures the worst overlap between any two components with respect to the kernel $\mathbf{k}$.

**Coupling parameter.** The coupling parameter is defined as:

$$\mathcal{C}(\mathbb{P}) := \max_m \left\| \frac{\mathbf{k}(\omega, \theta)}{q_m(\omega) q_m(\theta)} - w_m \frac{\mathbf{k}(\omega, \theta)}{q(\omega) q(\theta)} \right\|_{\mathbb{P}_m \otimes \mathbb{P}_m}^2, \tag{64}$$

where $q(\theta) = \sqrt{\int \mathbf{k}(\omega, \theta) \mathrm{d}\mathbb{P}(\omega)}$ and $q_m(\theta) = \sqrt{\int \mathbf{k}(\omega, \theta) \mathrm{d}\mathbb{P}_m(\omega)}$. It measures the coupling of function spaces over $\mathbb{P}_2$ with respect to the Laplacian operator. When it is $0$, for instance, the Laplacian over $\mathbb{P}$ is the weighted sum of Laplacians over $\mathbb{P}_m$ with weights $w_m$.

**Indivisibility parameter.** The indivisibility of a probability measure is defined as:

$$\Gamma(\mathbb{Q}) := \inf_{S \subset \Omega} \frac{p(\Omega) \int_S \int_{S^c} \mathbf{k}(\omega, \theta) \mathrm{d}\mathbb{Q}(\omega) \mathrm{d}\mathbb{Q}(\theta)}{p(S) p(S^c)}, \tag{65}$$

where $p(S) := \int_S \int_\Omega \mathbf{k}(\omega, \theta) \mathrm{d}\mathbb{Q}(\omega) \mathrm{d}\mathbb{Q}(\theta)$.

And $\Gamma_{\min}(\mathbb{P}) := \min_m \Gamma(\mathbb{P}_m)$ measures how easy it is to split a single component into two which is suggestive of ill-fittedness of the current model.

**Boundedness parameter.** Finally, we define:

$$b_{\max} := \max_m \left\| \frac{\mathbf{k}(\cdot, \theta)}{q_m(\cdot) q_m(\theta)} \mathrm{d}\mathbb{P}_m(\theta) \right\|_\infty^2. \tag{66}$$

This is just a constant when the kernel is bounded.

**The difficulty function.** With these parameters set up, we can now define the difficulty function used in Prop. 3:

$$\phi(\mathbb{P}, \mathbf{k}) := \frac{\sqrt{K(\mathcal{S}_{\max}(\mathbb{P}) + \mathcal{C}(\mathbb{P}))}}{\min_m w_m \Gamma_{\min}^2(\mathbb{P})}. \tag{67}$$

## F.2 FINITE-SAMPLE CONE STRUCTURE ELEMENTS

To get Theorem 2 from (Schiebinger et al., 2015), we require additional concepts and notations. For two vectors $u, v$ in $\mathbb{R}^K$, we define the angle between them $\mathrm{angle}(u, v) := \arccos \frac{\langle u, v \rangle}{\|u\| \|v\|}$. An orthogonal cone structure $OSC$ with parameter $\alpha, \theta$ is an embedding of $n$ points $\{(X_i \in \mathbb{R}^n, Z_i \in [K])\}_{i \in [n]}$ into $\mathbb{R}^K$ such that for each $m \in [K]$, we can find a subset $S_m$ with at least a $(1 - \alpha)$ proportion of all points with $Z_i = m$ where any $K$ points taken from each one of these subsets have pairwise angle at least $\theta$.

In the derivation of Thm. 6, Schiebinger et al. (2015) also let $b$ be such that $\mathbf{k} \in (0, b)$, and $r$ be such that $q_m(X^m) \geq r > 0$ with probability 1. $c_0, c_1, \ldots$ are then other constant that depends only on $b$ and $r$.

Table 3: Citation network details.

|          | Nodes ($N$) | Edges | Features | Classes ($C$) |
|----------|-------------|-------|----------|---------------|
| Cora     | 2708        | 10556 | 1433     | 7             |
| CiteSeer | 3327        | 9104  | 3703     | 6             |
| PubMed   | 19717       | 88648 | 500      | 3             |

In conjunction with other works on the topic, they also defined a tail decay parameter:

$$\psi(t) := \sum_{m=1}^{K} \mathbb{P}_m \left[ \frac{q_m^2(X)}{\|q_m\|_{\mathbb{P}}^2} < t \right] \tag{68}$$

and an extra requirement and the difficulty function: that there exists a $\delta > 0$ such that:

$$\phi(\mathbb{P}; K) + \frac{1}{\Gamma_{\min}^2(\mathbb{P})} \left( \frac{1}{\sqrt{n}} + \delta \right) \leq c\Gamma_{\min}^2(\mathbb{P}). \tag{69}$$

In words, it means that the indivisibility parameter of the mixture model is not too small relative to the clustering function. Finally, in the statement of Thm. 6, the difficulty parameter is reparameterized as the left hand side of Eq. (69):

$$\phi_n(t) := \phi(\mathbb{P}; k) + \frac{1}{\Gamma_{\min}^2(\mathbb{P})} \left( \frac{1}{\sqrt{n}} + \delta \right), \tag{70}$$

where $n$ is the number of points in the dataset.

## G ADDITIONAL EXPERIMENT DETAILS

All the code for the numerical experiments was written using the PyTorch and PyTorch Geometric libraries. The first set of experiments was run on an Intel i7 CPU, and the second set on an NVIDIA A6000 GPU.

**Transferability for node classification.** The details of the citation network datasets used in this experiment are displayed in Table 3. To perform graphon sampling, the nodes in these networks were sorted by degree. We considered a 60-20-20 training-validation-test random split of the data for each realization. In all scenarios, we trained a GNN consisting of a 2-layer GCN with embedding dimension 32 and ReLU nonlinearity, and 1 readout layer followed by softmax. We minimized the negative log-likelihood using ADAM with learning rate 0.001 and default forgetting factors over 100 training epochs.

**Positional encodings for graph classification.** We anonymized the MalNet-Tiny dataset by removing the node features and replacing them with the all-ones signal. Since we focus on large graphs, we further removed any graphs with less than 4500 nodes. This brought the number of classes down to 4, and we additionally removed samples from certain classes at random to balance the class sizes, yielding a dataset with 216 graphs in total (54 per class). We considered a 60-20-20 random split of the data for each realization. In all scenarios, we trained a GNN consisting of a 4-layer GCN with embedding dimension 64 and ReLU nonlinearity, and 1 readout layer with mean aggregation followed by softmax. We minimized the negative log-likelihood using ADAM with batch size 8, learning rate 0.001 and default forgetting factors over 150 training epochs. The PEs are the 10 first normalized Laplacian eigenvectors, and to obtain the graphon-sampled subgraph, we fix $\lambda = \lambda_{10}$, $q = 20$, $p = 10$, 2 communities, and $r = 10$.

