# OpenReview forum: "A Poincaré Inequality and Consistency Results for Signal Sampling on Large Graphs"
_ICLR.cc/2024/Conference — ICLR 2024 spotlight_

### Official Review · Reviewer_2prr · 2023-10-25

**Soundness:** 3 good
**Presentation:** 3 good
**Contribution:** 3 good
**Rating:** 8
**Confidence:** 4

**Summary:**

The authors propose a theory on signal sampling from graphons and provide a poincare type inequality for graphon signals. In particular, they show that unique sampling sets on a convergent graph sequence converge to unique sampling sets on the corresponding graphon. They propose a graphon signal sampling algorithm and empirically test its performance.

**Strengths:**

Originality and Novelty: The approach that the authors propose is, to the best of my knowledge, original and novel.
Significance: Nowadays it is certainly an interesting and important topics to study graphons, graph signal processing, and related tasks in ML. The machinery proposed in this paper, such as the graphon Laplacian and the graphon fourier transform, are themselves interesting constructs that could find wider adoption potentially.
Quality: The technical claims are, to the best of my knowledge, sound and reasonable, although I did not check any proofs in the appendix in detail. The convergence results on uniqueness sets are sufficiently deep for publication at a venue like ICLR.
Clarity: The article is written moderately clearly, but is presented in a very dense manner.

**Weaknesses:**

The main weakness is in the presentation and exposition:

The paper is presented in a very dense way,  My main suggestion to the authors is to provide more motivation and background for the readers. For example, a general ML audience might not have a very good idea of what a poincare inequality is. Setting the motivation and background up more allows for a broader audience and a more enjoyable read.

**Questions:**

- One canonical way to motivate graphons is via the Aldous-Hoover theorem, which motivates a latent variable perspective on graphons if one sees the nodes as exchangeable. I think the paper by Aldous (1981) and Hoover (1979) are classical references that should be cited/acknowledged in studies discussing graphons.

- One potential limitation of the graphon sampling approach is that many real life graphs are sparse, but graphons can only be reasonably motivated by dense graphs. Obviously, this is not a short coming of this research (since it is an open problem), but I think it would be reasonable for the authors to mention this explicitly in their paper to provide a comprehensive/balanced perspective.

---

> ### Author Response · Authors · 2023-11-21
> **Response to Reviewer 2prr**
>
> We thank the reviewer for their time and thoughtful comments and will now address the questions raised.
>
> 1. **Density of the paper and more motivation and reference needed.**
> We thank the reviewer for their patience and would like to reassure them that we are working to make the paper more readable for the general ML audience. We have included a further explanation of Poincaré inequality in this setting, as well as Aldous and Hoover’s reference, which we agree is also a natural motivation for graphons, especially if one is to extend this framework to sparser limits.
>
> 2. **Sparse graphs.**
> - We completely agree with the reviewer that developing similar frameworks for sparse graph limits is tremendously useful but at the same time tremendously difficult. In particular, most notions of sparse graph limits do not admit continuous eigenvalues/eigengaps at the limit. For some pathological cases, eigenvalues may not even exist for the limit object! As our framework is built on Fourier transform of the Hilbert-Schmidt operator of graphon, the degeneracy in sparse limit spectrum simply means that a lot more care is needed to define a unitary transform for sparse graph limits.
> - Our results, however, also shed some positive light in the sparse case: in our proposition 4, after a large enough number of node N, assuming the generating graphon model admits nice mixture model reparameterization, the size of the uniqueness set no longer increases with the number of nodes n > N. This suggests that even in the limit of infinitely many nodes, there may still be a finite uniqueness set, which might be possible to be picked out by a more fine-grained topology than that offered by the Lebesgue measure!

---

### Official Review · Reviewer_3x8Y · 2023-10-30

**Soundness:** 4 excellent
**Presentation:** 3 good
**Contribution:** 3 good
**Rating:** 8
**Confidence:** 2

**Summary:**

This paper studies graph reductions from the perspective of graphons. It finds a subset of vertices such as a signal on them can be uniquely reconstructed on the rest of the graph, given that the signal comes from a limited family. When the family of signals are coming from the top few eigenvalues, it gives an algorithms based on sampling points from intervals that form graphons. The effectiveness of this method is then experimentally tested against node classification and eigenvalues.

**Strengths:**

The paper is closely connected with the extensive literature on graphons, as well as prior works on graph size reductions. It gives both theoretical bounds as well as experimental verifications of the approaches.

**Weaknesses:**

The algorithms as well as theorems in the paper feel more motivated by the theory of graphons than concrete applications. While there is a lot of general utility in such work, the amount of definitional work feels overwhelming to me, especially given that the end goal is to preserve eigenvalues / quality of classification algorithms. However, this is understandable given the extensive depth of this topic.

**Questions:**

Does the theorems developed in Section 4 imply a direct bound on the quality of applying the algorithm in Section 5 to node classification?

---

> ### Author Response · Authors · 2023-11-21
> **Response to Reviewer 3x8Y**
>
> We thank the reviewer for their time and thoughtful comments and will now address the questions raised.
> 1. **Heavy notations/definitions/theorem statements.**
> We have and will continue to work to simplify some of the notations in the main paper. For a simplified setting of stochastic block models where these notions are not necessary, please refer to Proposition 5 in Appendix D for a simplified exposition. To state a more general theorem for more complicated and realistic graphon models, we unfortunately need to build on foundational notions uncovered by Schiebinger et al. in their seminal paper in 2015. We have included all relevant definitions and intuitions for these definitions in Appendix F.
>
> 2. **The algorithm was stated rather informally and whether the theorems in Section 4 imply a guarantee for the stated algorithm for node classification.**
> - When the conditions stated in theorem 3-5 are met, these theoretical guarantees do apply to the algorithm stated in section 5, provided that in step 2 of the algorithm, we use Gaussian elimination. This is formalized in Proposition 4 where with a large enough n, uniqueness set of G_n are clean samples from each mixture component (Theorem 4), which can be found via Gaussian elimination (Theorem 5). We have stated this more clearly in the main paper.
> - However, it should also be noted that in order to make our algorithm even faster and competitive with existing finite graph sampling methods such as Anis et al. 2016, we use the same greedy heuristic as Anis et al., in place of Gaussian elimination in our experiments, as discussed in the paragraph following the algorithm. Our proposed algorithm is also highly flexible in practice and is amenable to more sophisticated sampling tools, such as via a local clustering algorithm in step (3).  For these more sophisticated add-ons, theoretical guarantees are left open. However, we believe that the demonstrated flexibility is extremely important in making sure that our framework is useful in practice, while still being built on solid theoretical foundations.
>
> Reference: A. Anis, A. Gadde, and A. Ortega. Efficient sampling set selection for bandlimited graph signals using graph spectral proxies. IEEE Trans. Signal Process., 2016.

---

> > ### Comment · Reviewer_3x8Y · 2023-12-04
> >
> > Thank you for answering my questions and concerns, and for pointing me to the more extensive literature. I will raise my score based on these further explanations.

---

### Official Review · Reviewer_1oL4 · 2023-10-31

**Soundness:** 3 good
**Presentation:** 4 excellent
**Contribution:** 4 excellent
**Rating:** 8
**Confidence:** 4

**Summary:**

This work considers the extension of sampling theory for bandlimited graph signals to signals on graphons, generalizing the notion of a finite sampling set for graphs to a measurable sampling set for graphons. With this notion in hand, the authors propose an algorithm for consistent sampling, as well as demonstrate its utility in training and inference with GNNs.

**Strengths:**

1. This paper reads very well: relevant background, theory, and results are presented clearly and in a logical manner. I can tell that the authors put a lot of thought into how this paper should be presented, which is very much appreciated.
2. The results offered by the authors are novel and interesting, while still fitting well with the literature on graph/graphon signal processing.
3. In studying this problem, the authors do not restrict to one single approach. Rather, they show two different solutions to Problem 2 via different methods, which makes for a very good discussion.
4. In Section 5, the authors make a concrete connection between their theory of graphon uniqueness sets and applications to large graphs.

**Weaknesses:**

1. Applying knowledge of the uniqueness set for a graphon to graphs sampled from that graphon is difficult without knowing about the latent position of the graph's nodes. The authors acknowledge this weakness at the end of Section 5, though, so I don't count this as too strong of a weakness.
2. It is not clear how robust the constructed uniqueness sets are to violations of perfect bandlimitedness. For instance, if a signal is only approximately bandlimited w.r.t. some cutoff frequency $\lambda$, I would hope that a reconstruction of that signal using a uniqueness set for $\lambda$ would be stable.

**Questions:**

1. In the statement of Theorem 2, signals $X\in L^2(S)$ are considered, but then the operator $L$ is applied to $X$. It is not clear here which Laplacian $L$ is: I presume it is the scaled normalized Laplacian of $\Gamma(S)$. My question, then, is how is the Laplacian understood to act on $X$ when $X$ is not a signal in $L^2(D)$, but rather $L^2(S)$. Is there a canonical inclusion $L^2(S)$ in $L^2(D)$ that I am missing here? Please clarify the statement of Theorem 2. Could you please provide a comment on Point 2 in the weaknesses section above? In particular, I would like to see further justification for applying methods based on signal bandlimitedness to the datasets used in Section 6. It is not obvious to me if these signals are bandlimited, in which case I have no reason to believe that the uniqueness sets yield stable representations of the signals.

---

> ### Author Response · Authors · 2023-11-21
> **Response to Reviewer 1oL4**
>
> We thank the reviewer for their time and thoughtful comments and will now address the questions raised.
> 1. **Theorem 2 (Poincaré inequality)**
> We thank the reviewer for pointing out a missing definition! Indeed the reviewer is correct that there is a canonical inclusion of $L_2(S)$ in $L_2([0,1])$ when $S$ is a subset of $[0,1]$: a graphon signal $X$ is in $L_2(S)$ if it is in $L_2([0,1])$ and the support of $X$ is in $S$. In other words, $X$ is $0$ everywhere outside $S$. The short proof of Theorem 3 details the intuition for this definition. But at a high level, we want to show that $S$ is not informative by showing that any bandlimited signal $X, Y$, with $X - Y$ supported only on $S$ (i.e. $X$ in $L_2(S)$), must have ${\|| X -  Y \||}\_{L_2} = 0$. In other words, annihilating all parts of the signals from $[0,1]\S$ makes these signals indistinguishable (in $L_2$). To show this, we upper bound $||X - Y||\_{L_2}$. This is done via the Poincare inequality: $||X - Y||\_{L_2} \leq {\rm constant} ||\mathcal{L}(X - Y)||\_{L_2}$. The right hand side is then at most $\lambda ||X - Y||\_{L_2}$ due to bandlimitedness of $X$ and $Y$.  $||X - Y||\_{L_2} = 0$ quickly follows.
> 2. **Robustness to perturbation**
> - We agree that a perturbation theoretic analysis into the construction of these uniqueness sets is a very important and interesting future direction. In short, it is reasonable to expect the graphon uniqueness set to be robust to perturbation since Theorem 1 and Proposition 2 connect uniqueness sets to full-rank systems, and full-rankedness is a robust property: a matrix is full-rank iff its determinant is nonzero, so a small enough perturbation on the eigenvalues of the system will not cause the determinant to become 0. In fact, much of the technical part of the proof uses Gershgorin circle theorem to show full-rankedness of different systems!.
> - To give another intuition, by Proposition 4, if the limiting graphon admits a nice (small difficulty function) mixture model reparameterization, we have that after a finite number of nodes $N$, uniqueness set of the finite graph $G_n$ does not change for all $n > N$. This implies that in the limit of infinitely many nodes, there is a uniqueness set with discretely many nodes. The theory of graphon is not enough to capture this fine-grained limit since graphons are $L_2$ functions and are only defined up to a set of measure 0. However, any measurable subset (let’s say union of intervals in general positions) containing these N points should contain enough information to be a uniqueness set themselves. Thus, a small perturbation of the boundaries of these intervals would not affect uniqueness of the computed graphon uniqueness set unless one of these N points lies exactly on a boundary of an interval, which is unlikely. In practice, q - the size of the uniqueness set, is a tunable hyperparameter that practitioners can control to obtain a tradeoff between compute-saving and information loss, in a case where bandwidth limit is not known a priori.

---

### Official Review · Reviewer_KQ8B · 2023-11-06

**Soundness:** 3 good
**Presentation:** 3 good
**Contribution:** 3 good
**Rating:** 6
**Confidence:** 3

**Summary:**

This paper proposes a novel signal sampling theory on a type of graph limit called a graphon. The authors demonstrate a Poincaré inequality for graphon signals and show that the complements of node subsets satisfying this inequality are unique sampling sets for Paley-Wiener spaces of graphon signals. They leverage connections with spectral clustering and Gaussian elimination to prove the consistency of these sampling sets. They also design a graphon signal sampling algorithm for large graphs and validate its performance on graph machine learning tasks.

**Strengths:**

1. The paper introduces an innovative approach to signal sampling on large-scale graphs using graphons, a type of graph limit, making it a significant contribution to the field.

2. The authors successfully prove a Poincaré inequality for graphon signals and demonstrate that the complements of node subsets adhering to this inequality form unique sampling sets. These solid theoretical foundations enhance the reliability of their proposed methodology.

3. The authors make a critical contribution by relating bandlimitedness in graphon signal space to optimal sampling sets. This achievement not only generalizes previous findings on finite graphs but also offers a new perspective towards problem-solving in the field.

**Weaknesses:**

The paper does not compare the computational efficiency of the proposed method with existing techniques in the field. Such a comparison could show whether the new method offers improvements in terms of time efficiency.

**Questions:**

1. Given that the paper focuses on large graphs, how does the proposed graphon signal sampling algorithm scale with the size of the graph? While the paper discusses theoretical aspects and empirical performance, it does not delve into scalability and computational efficiency. Can the authors provide insights or discussions on how their algorithm performs as the graph size increases?

2. The authors have demonstrated the utility of their method on specific tasks. However, how broadly applicable is the proposed method across different graph machine learning tasks? Can the authors provide examples or discussions on how their method could be applied to other types of problems within the graph machine learning domain?

---

> ### Author Response · Authors · 2023-11-21
> **Our response to Reviewer KQ8B**
>
> We thank the reviewer for their time and thoughtful comments and will now address the questions raised
>
> 1. **Compare computational efficiency with existing techniques, and how does the proposed algorithm scale with the size of the graph?**
> - In the last paragraph of Section 5 on page 8, we provided a runtime analysis of the main computational step (step 2) of our sampling algorithm compared to the greedy heuristics in Anis et al., 2016. We have made this comparison more explicit in the main paper. The specific runtime of Anis et al. 2016 has been carefully analyzed and compared with other methods in their paper, and we will include these findings in the Appendix for completeness, together with our own numerical runtime.
> - While we do gain some efficiency in step (2) of our algorithm compared to existing graph sampling algorithms, the main advantage and efficiency of our proposed method come from not having to redo this sampling step every time the graph grows in size. This is due to the fact that the sampled structures follow the same graphon. We also experience significant savings in downstream analysis tasks. For instance, training GNNs on smaller graphs (of size q << n) and transferring to testing on larger graphs, or computing positional encoding of size q and then 0-padding to get positional encoding of size n, all benefit from massive savings since the learning algorithms only interact with these smaller graphs.
> - In real-world datasets, we can expect q to be much less than n, as the uniqueness set captures some notion of the intrinsic difficulty of the problem while n captures the resolution of the problem. This intuition is rigorously supported by our theorems, which demonstrate that q scales with the number of mixture components/number of communities when the underlying limit graphon can be parameterized as a mixture model with a low difficulty function value. In a simple setting of stochastic block models, such as Proposition 5 in Appendix D, it is clear that q is, in fact, independent of n, and our algorithm does not become slower as n increases (ignoring the preprocessing steps (1) and (3)).
>
> 2. **Provide examples of other applications within the graph machine learning domain.**
> The paper proposes the use of information contained in a certain small subset of nodes (a uniqueness set) to solve problems on the full graph.
> - Practically, our algorithm allows q - the size of the subset - to be a tunable hyperparameter for practitioners to choose and obtain an informative subset of size q. Apart from our citation network experiments involving graph classification and positional encoding using almost eigenvectors, there exist numerous other, more nuanced tasks where focusing on this small subset could be advantageous. However, we have not attempted these and leave them for future studies. For example, in a scenario with a vast unlabeled dataset with graph-based input, one could imagine finding a crucial, small subset for manual labeling and subsequently learning an algorithm to predict other labels in a semi-supervised manner. In another scenario, consider a dataset with censors as nodes in a graph. Applying our algorithm to identify a key subset of censors to focus computational efforts on is a feasible and scalable idea.
> - Theoretically, our framework works as long as one can justify a low signal bandwidth setting for the task at hand. Under the lens of mixture models, one way to justify such low-bandwidth assumptions, as we have demonstrated, is when there is a process that aggregates nodes into a small number of clusters, leading to regularity in the frequency domain.
>
> Reference:
> A. Anis, A. Gadde, and A. Ortega. Efficient sampling set selection for bandlimited graph signals using graph spectral proxies. IEEE Trans. Signal Process., 2016.

---

> > ### Comment · Reviewer_KQ8B · 2023-11-22
> >
> > Thank you for your response.
> >
> > However, I find that my concerns regarding the computational efficiency and scalability of the proposed method have not been addressed well. The absence of either a theoretical computational complexity analysis or an empirical numerical simulation analysis fails to illustrate the scalability of your approach. This may lead me to adjust my rating downward.

---

> > > ### Author Response · Authors · 2023-11-22
> > > **Author response #2**
> > >
> > > We thank the reviewer for the prompt reply and invite them to review the latest version of the manuscript in which we have added:
> > > 1. Additional columns to Table 1, comparing **classification runtime** in seconds between experiments using full graph/graphon sampled nodes (our proposed method) and uniformly randomly sampled nodes.
> > > 2. Additional columns to Table 2, comparing **PE computation time** in seconds between experiments using full graph and  sampled subgraphs.
> > > In both of these analyses, running the downstream task on sampled nodes greatly reduces runtime.
> > > 3. Additionally, we have added a specific **time complexity analysis** of sampling in our framework compared to Anis et al. 2016 in the paragraph 'Runtime analysis' in page 8. We are hiding factors that depend on specific hyperparameter choices of the heuristic, which is identical between ours and Anis et al. 2016. In particular, step 1 of our algorithm costs $O(|E|)$ time to parse the input graph, step 2 costs $O(pq^2)$ time to subsample and step 3 costs $O(m)$ time (if we use uniform samples instead of sophisticated heuristics such as local clustering). In comparison, Anis et al. 2016 requires $O(m|E|)$ time to run step 2 on the full graph and obtain uniqueness set of size $m$. Note that $|E|$ can be up to $n^2$ and thus much larger than $q^2$ (conversely, $q$ may not rely on $n$ at all), and $m$ is always larger than $p$ ($m \approx rp$, where $r$ is a hyperparameter in our algorithm).

---

> > > > ### Comment · Reviewer_KQ8B · 2023-11-22
> > > >
> > > > Thank you for your detailed response. My concerns have been addressed well. I would suggest adding comments on the runtime of the numerical simulations in Section 6 for further clarity. I will maintain the current rating without any downward adjustments. I look forward to seeing the final version.

---

### Meta-Review · Area_Chair_ogHD · 2023-12-05

**Metareview:**

This article introduces a novel approach for signal sampling on graphs, based on graphon graph limit theory. The authors prove a Poincaré inequality for graphon signals, and show that complements of node subsets satisfying this inequality are unique sampling sets for Paley-Wiener spaces of graphon signals. The authors also derive a graphon signal samplign algorithm for large graphs, and provide experiments on graph machine learning tasks.
The paper provides an innovative approach to this problem, and brings a number of novel theoretical contributions. The exposition is clear but rather dense, which is somewhat expected as it covers a lot of different topics. Based on the reviews and the author's feedback, I recommend the paper to be accepted.

**Justification For Why Not Higher Score:**

The topic may be somewhat a bit niche for the ICLR audience.

**Justification For Why Not Lower Score:**

This article introduces a novel approach for signal sampling on graphs, based on graphon graph limit theory.

---

### Decision · Program_Chairs · 2024-01-16

Accept (spotlight)